# StreamBridge: Turning Your Offline Video Large Language Model into a Proactive Streaming Assistant

**Haibo Wang**[1,2][*] **Bo Feng**[1°]**, Zhengfeng Lai**[1°]**, Mingze Xu**[1°]**, Shiyu Li**[1°]**,**
**Weifeng Ge**[2]**, Afshin Dehghan**[1†]**, Meng Cao**[1†]**, Ping Huang**[1†]**,**
[1]Apple   [2]Fudan University
hibwang@ucdavis.edu
{bfeng2, jeff_lai, mingze_xu2, shiyu_li}@apple.com
{adehghan, mengcao, huang_ping}@apple.com
[*]First author; [°]Core contributors; [†]Senior authors

## Abstract

We present ***StreamBridge***, a simple yet effective framework that seamlessly transforms offline Video-LLMs into streaming-capable models. It addresses two fundamental challenges in adapting existing models into online scenarios: (1) limited capability for multi-turn real-time understanding, and (2) lack of proactive response mechanisms. Specifically, *StreamBridge* incorporates (1) a memory buffer combined with a round-decayed compression strategy, supporting long-context multi-turn interactions, and (2) a decoupled, lightweight activation model that can be effortlessly integrated into existing Video-LLMs, enabling continuous proactive responses. To further support *StreamBridge*, we construct ***Stream-IT***, a large-scale dataset tailored for streaming video understanding, featuring interleaved video-text sequences and diverse instruction formats. Extensive experiments show that *StreamBridge* significantly improves the streaming understanding capabilities of offline Video-LLMs across various tasks, outperforming even proprietary models such as GPT-4o and Gemini 1.5 Pro. Simultaneously, it achieves competitive or superior performance on standard video understanding benchmarks.

## 1   Introduction

Video Large Language Models (Video-LLMs) [1; 2; 3; 4; 5] typically process entire pre-recorded videos at once. However, emerging applications, such as robotics [6; 7] and autonomous driving [8; 9], require causal perception and interpretation of visual information online. This fundamental mismatch highlights a critical limitation of current Video-LLMs, as they are not inherently equipped to operate in streaming scenarios where timely understanding and responsiveness are paramount.

Figure 1 highlights two representative patterns in streaming video understanding, which also correspond to the key challenges in adapting Video-LLMs from offline to streaming scenarios: *(1) multi-turn real-time understanding* and *(2) proactive response generation*. The first pattern involves multi-turn interactions, where the assistant receives user queries at different timestamps. In each turn, while keeping accumulated visual and conversational context as historical information, the model should focus on the most recent video segment. The second pattern emphasizes more human-like, proactive behaviors. Rather than passively waiting for user prompts, the model actively monitors the visual stream and generates timely outputs based on unfolding content. For instance, in Figure 1 (bottom), the assistant provides step-by-step guidance as the drawing progresses without being explicitly asked, simulating continuous support in dynamic environments.

---

[*]Work done during Haibo's internship at Apple.

39th Conference on Neural Information Processing Systems (NeurIPS 2025).

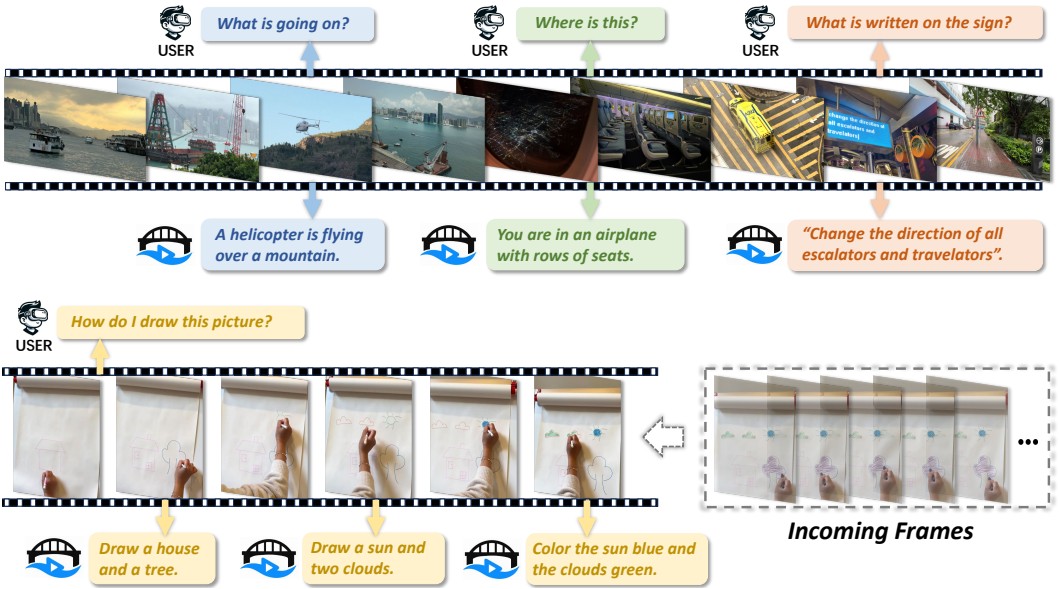

Figure 1: Illustration of streaming scenarios. Top: Multi-turn interactions. User issues queries at different timestamps, with each turn involving a new video segment along with accumulated visual and text history. Bottom: Proactive responses. The assistant actively delivers timely feedback or guidance based on the incoming visual stream, without requiring explicit user prompts.

To bridge the gap between offline and streaming video understanding, we introduce **StreamBridge**, a simple yet effective framework that seamlessly transforms pre-trained offline Video-LLMs into streaming-capable models. In contrast to prior efforts [10; 11; 12; 13], which train streaming models from scratch but fall behind on offline video tasks, *StreamBridge* leverages the strong generalization capabilities of existing Video-LLMs without requiring full retraining. This approach allows developers to directly benefit from the rich world knowledge and linguistic fluency of large-scale pre-trained models, while incurring only minimal additional computational cost and data requirements. Concretely, *StreamBridge* introduces a memory buffer to manage incoming video frames, coupled with a round-decayed compression strategy that merges earlier frame tokens while preserving recent ones, enabling the model to support long-context, multi-modal, and multi-turn interactions in streaming scenarios. For proactive capabilities, instead of modifying the base model architecture [14] or introducing streaming-specific objectives [12], both of which can lead to optimization conflicts and issues like probability correction [10], *StreamBridge* adopts a modular design, by decoupling the proactive capability from the main Video-LLM via a compact activation model. This plug-and-play component operates in parallel with the main Video-LLM, enabling proactive behavior in a flexible and non-intrusive manner while fully preserving the main Video-LLM's language fluency and general video understanding capabilities.

To further support *StreamBridge*, we construct **Stream-IT**, a large-scale dataset tailored for streaming scenarios. *Stream-IT* captures diverse real-time questions and proactive responses embedded within multi-turn video interactions, featuring interleaved video-text sequences. While existing datasets primarily focus on single-turn question answering [15; 16] or short-form video captioning [17; 18; 19], *Stream-IT* fills a critical gap by enabling temporally extended, interactive video understanding. It is constructed by concatenating semantically related short clips from large-scale video-caption corpora, followed by the generation of multi-turn QA sequences that simulate realistic, time-sensitive user interactions. Moreover, *Stream-IT* incorporates a broad spectrum of task formats sourced from public datasets, thereby boosting task diversity and promoting model generalization in streaming settings.

By integrating our *StreamBridge* framework and fine-tuning on *Stream-IT*, we successfully convert several leading offline Video-LLMs, including LLaVA-OV [3], Oryx-1.5 [1], and Qwen2-VL [2], into streaming-capable assistants. Extensive experiments demonstrate that our models achieve state-of-the-art performance on streaming benchmarks such as OVO-Bench [20] and Streaming-Bench [21], outperforming even proprietary models like GPT-4o [22] and Gemini 1.5 Pro [23], while retaining or exceeding performance on conventional offline video understanding tasks [24; 25; 26; 27; 28; 29].

## 2   Related Work

**Video Large Language Models.** With the rapid advancement of Multimodal Large Language Models (MLLMs) [30; 3; 31; 32; 2], Video-LLMs [33; 34; 35; 36; 15; 37; 38] have gained increasing attention for general video understanding. Typically, these models comprise a visual encoder [39; 40; 1] for extracting frame-level representations, a modality projector (*e.g.,* MLP [41] and Q-former [30]) to map visual features into the language space, and an LLM [42; 43] to generate contextual responses. While achieving strong results on standard video benchmarks [25; 29; 27], these models are inherently designed for static, offline settings where the entire video is pre-recorded and fully accessible at inference time. As a result, they struggle in streaming environments, where video frames arrive sequentially and require real-time, temporally coherent, or even proactive responses. Our work aims to bridge this gap by augmenting offline Video-LLMs with streaming capabilities.

**Streaming Video Understanding.** Typical tasks in streaming video understanding, such as action recognition [44; 45; 46; 47] and anticipation [48; 49], causally process video inputs using only past and current observations. Recent efforts [50; 12; 13; 51] focus on building Video-LLMs capable of real-time conversation, generating timely responses throughout a live video stream. VideoLLM-Online [10] and Flash-VStream [11] introduce specialized online objectives and memory architectures to handle sequential inputs. MMDuet [14] and ViSpeak [52] add dedicated heads to facilitate proactive response generation. To benchmark streaming video capabilities, several evaluation suites have been proposed, including StreamingBench [21], StreamBench [53], SVBench [54], OmniMMI [55], and OVO-Bench [20]. In contrast to previous approaches that retrain models or tightly couple proactive mechanisms within the backbone, our work leverages the strong generalization abilities of pre-trained offline Video-LLMs [1; 3; 2]. We propose an efficient adaptation framework, combined with a dedicated fine-tuning dataset, to endow these models with streaming capabilities. Furthermore, observing that embedding the activation function into the main model often lead to optimization conflicts and performance degradation [10; 14], we advocate a modular, decoupled design. Our method introduces a compact, plug-and-play activation model that enables proactive behaviors efficiently and non-intrusively. We also provide additional discussions on how our work relates to the ReKV [56], VideoStreaming [57], and StreamChat [53] in the Appendix F.

## 3   Methodology

### 3.1   Preliminary Analysis

Streaming video understanding involves interleaved video-text inputs. From an input perspective, streaming scenarios can be broadly categorized into two representative formats:

- **Multi-turn dialogue with interleaved video-text.** In this setting, the input sequence is in the form of '$<V_1> <Q_1> <A_1>$, $<V_2> <Q_2> <A_2>$, $\cdots$', where $<V_i>$, $<Q_i>$, and $<A_i>$ denote the video clip, user query, and assistant answer in the $i$-th round. Crucially, there is no delay between $<Q_i>$ and $<A_i>$, reflecting the need for immediate responses. This format closely resembles the live interaction in dynamic environments, as shown in Figure 1 (Top).

- **Proactive output.** The assistant answers *after* watching an incoming video stream, often without an explicit user query at the response time. The input can be structured as '$<Q> <V_1> <A_1> <V_2> <A_2> \cdots$', where $<Q>$ represents an initial prompt (*e.g.,* "Guide me through the task"), and the model must proactively determine when and how to respond based on the incoming video contents. This scenario requires the ability to continuously monitor evolving context and trigger responses at appropriate moments. Figure 1 (Bottom) is an example of proactive responses.

Recent benchmarks such as OVO-Bench [20] and Streaming-Bench [21] attempt to evaluate these capabilities by constructing multi-turn interleaved video-text dialogues. However, due to the limited input length and the lack of streaming support in current Video-LLMs, these benchmarks necessarily simplify the problem. Specifically, they segment a complete long video into multiple isolated clips aligned with each query timestamp. For a query $<Q_i>$ at time $t_i$, the visual input is restricted to the uniformly sampled frames under segment $V_{[0:t_i]}$, and prior dialogue history is completely discarded. As a result, the multi-turn streaming scenario is reduced to a series of independent, single-turn offline tasks. To address these limitations, we propose *StreamBridge*, a general framework designed to introduce the actual streaming setup to existing offline Video-LLMs.

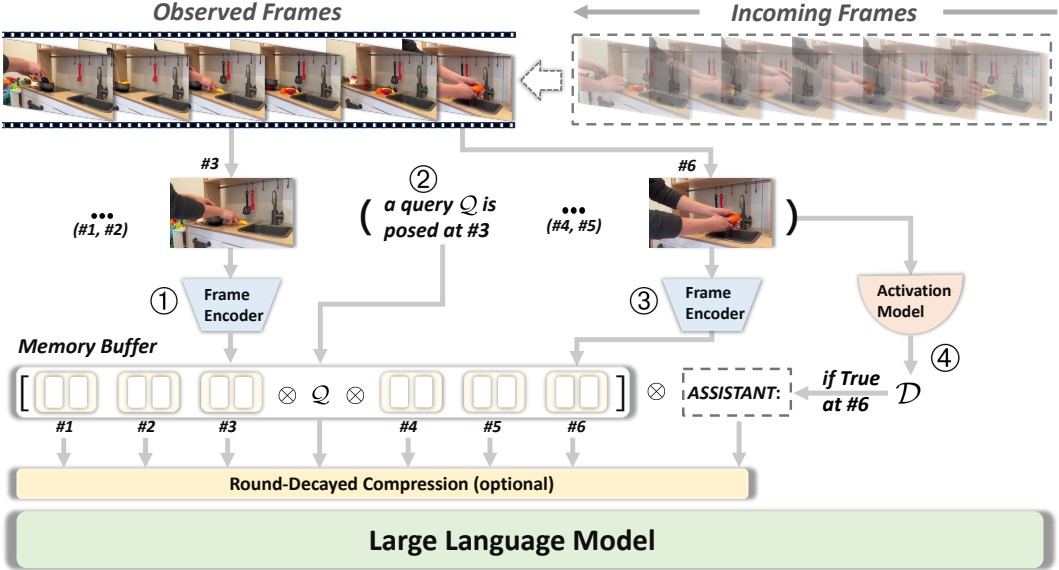

Figure 2: Overview of *StreamBridge*. ①③: Incoming frames are encoded and stored into the memory buffer one by one. ②: A query $\mathcal{Q}$ is posed. ④: The activation model monitors incoming frames and returns a binary signal $\mathcal{D}$, indicating whether LLM should start answering. $\otimes$ means concatenation.

## 3.2 StreamBridge

As shown in Figure 2 and Algorithm 1, in addition to the frame encoder $\mathcal{I}(\cdot)$ and the large language model $\mathcal{LLM}(\cdot)$, *StreamBridge* proposes three key components to enable streaming capabilities: (1) a memory buffer responsible for storing and retrieving frame tokens over time, (2) a round-decayed compression strategy $\mathcal{COM}(\cdot)$ that efficiently prunes redundant tokens from earlier rounds while preserving the most recent context, and (3) a compact activation model $\mathcal{ACT}(\cdot)$ that enables proactive responses by making frame-level decisions on when to generate outputs.

### 3.2.1 Memory Buffer

In streaming scenarios where video frames arrive sequentially, we adopt a memory buffer $\mathcal{MB}$ to store both visual and textual embeddings. As illustrated in Figure 2, each incoming frame is independently encoded and appended to the buffer alongside any associated query embeddings. Conceptually, $\mathcal{MB}$ operates under a producer-consumer paradigm: the encoder $\mathcal{I}(\cdot)$ functions as the producer, continuously generating frame-level features, while the language model $\mathcal{LLM}(\cdot)$ serves as the consumer, retrieving the accumulated embeddings to generate a response upon receiving a user query. Formally, as detailed in Algorithm 1, at each time step $t$, the incoming frame $F_t$ is first processed by $\mathcal{I}(\cdot)$, and the resulting embeddings are stored in the memory buffer $\mathcal{MB}$ (Algorithm 1, line 4). Upon the arrival of a user query $\mathcal{Q}$ and a positive activation decision $\mathcal{D}$, the buffer content, including both visual and textual embeddings, is flattened into a single sequence of input embeddings, which is then fed into $\mathcal{LLM}(\cdot)$ for response generation (Algorithm 1, line 13-16). Once a response $\mathcal{R}$ is produced, it is also appended to the memory buffer (Algorithm 1, line 17), enabling the model to preserve temporal continuity and maintain a complete history of multi-turn video-text interactions.

### 3.2.2 Round-Decayed Compression

Online scenarios often involve long, even infinite video streaming, which can lead to significant memory usage and inference latency. Therefore, we propose a round-decayed token compression strategy tailored for multi-turn streaming settings. Specifically, we pre-define a maximum allowable embedding length $\text{MaxLen}$ for the model input. Before each response generation, the model checks whether the current input embedding exceeds $\text{MaxLen}$. If so, we apply a round-decayed token merging strategy: starting from the earliest dialogue rounds, visual tokens are progressively merged frame-by-frame, until the total length falls below $\text{MaxLen}$. The merging is implemented via average

---

**Algorithm 1:** StreamBridge Framework

---

1    **Inputs:** incoming frames $[F_1, F_2, \ldots, F_t, \ldots]$;
2    **Initializations:** $\mathcal{I}(\cdot), \mathcal{LLM}(\cdot), \mathcal{ACT}(\cdot), \mathcal{COM}(\cdot), \mathcal{MB} = [\cdot], \text{MaxLen}, t_{\mathcal{Q}}=\text{None}$;
3    **while** $F_t$ **do**
4       $\mathcal{MB} \leftarrow \mathcal{I}(F_t)$ ;                    // store the frame feature $\mathcal{I}(F_t)$ into the Memory Buffer
5       **if** $\mathcal{Q}$ *at* timestamp $t$ **then**
6         $\mathcal{MB} \leftarrow \mathcal{Q}$
7         $t_{\mathcal{Q}} \leftarrow t$ ;                   // $t_{\mathcal{Q}}$ is the timestamp when $\mathcal{Q}$ is posed
8       **if** $t_{\mathcal{Q}}$ is not None **then**
9         $\mathcal{D} \leftarrow \mathcal{ACT}(\mathcal{Q}, F_{t_{\mathcal{Q}}:t-1}, F_t)$ ;      // $\mathcal{D}$ denotes whether response or not at timestamp $t$
10      **else**
11        $\mathcal{D} \leftarrow \text{False}$ ;                 // not response if there is no $\mathcal{Q}$
12      **if** $\mathcal{D}$ **then**
        // $\mathcal{D}$ is true at timestamp $t$, and should return a response $\mathcal{R}$
13        InputEmbeds $\leftarrow$ Flatten($\mathcal{MB}$)
14        **if** Len(InputEmbeds) $>$ MaxLen **then**
15          InputEmbeds $\leftarrow \mathcal{COM}$(InputEmbeds) ;      // compress redundant visual tokens
16        $\mathcal{R} \leftarrow \mathcal{LLM}$(InputEmbeds) ;            // return a response $\mathcal{R}$
17        $\mathcal{MB} \leftarrow \mathcal{R}$ ;                   // update $\mathcal{MB}$
18      $t \mathrel{+}= 1$ ;                      // receive subsequent frames

---

pooling [58] over adjacent frame tokens. This strategy ensures that the most recent visual context is retained with minimal distortion, thus maintaining the precision of real-time responses while not fully discarding historical visual contexts. At the same time, it significantly improves memory efficiency and reduces inference overhead as in Figure 4. This process is encapsulated in the compression function $\mathcal{COM}(\cdot)$ in Algorithm 1 (line 15). The detailed pseudo codes can be found in Appendix I.

### 3.2.3   A Plug-and-play Activation Model

To enable proactive responses in streaming Video-LLMs, we decouple the activation function from the main Video-LLM. Unlike prior methods that tightly integrate activation mechanisms into the LLM [10; 12; 14; 52], our framework avoids potential interference with the language modeling capacity of the main

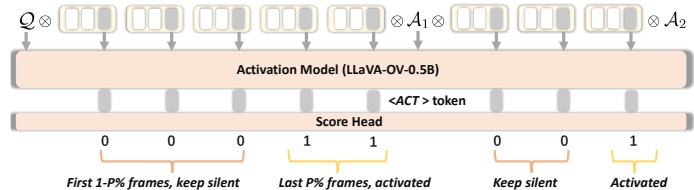

Figure 3: An overview of the proposed activation model. We label the last $P\%$ of frames of each video clip to be true during training.

Video-LLM. Specifically, we propose a parallel pipeline, where a compact external MLLM (*e.g.,* LLaVA-OV-0.5B [3]) is used as an independent activation model, denoted as $\mathcal{ACT}(\cdot)$. As shown in Algorithm 1 (line 9), upon receiving each new frame, the framework simultaneously forwards the current frame (along with the user query $\mathcal{Q}$ and optionally previous frames) to $\mathcal{ACT}(\cdot)$ to determine whether a response should be generated. If the activation signal $\mathcal{D}$ is positive, the buffered embeddings are sent to the LLM for decoding. This design ensures high flexibility and compatibility. Furthermore, in real-time deployment, the $\mathcal{ACT}(\cdot)$, the frame encoder $\mathcal{I}(\cdot)$, and the main $\mathcal{LLM}(\cdot)$ can run concurrently in parallel threads, enabling more efficient inference.

To train the activation model (illustrated in Figure 3), we modify the architecture by replacing the standard language modeling head with a score head for binary classification, and introduce a learnable activation token $<ACT>$ which is appended to the visual embeddings of each frame. After processing through the final layer, we extract the latest frame's activation token and feed its hidden representation into the score head to predict whether the model should respond at that time. During inference, only when the predicted score is greater than the activation threshold $\alpha$, the main Video-LLM can be triggered to give a response. Since $\mathcal{ACT}(\cdot)$ performs only binary classification (*i.e.,* to respond or not), we aggressively pool its visual tokens for efficiency. The input sequence to the model follows the format: '$<Q>$ $<V_1>$ $<A_1>$ $<V_2>$ $<A_2>$ $\cdots$', where the question $\mathcal{Q}$ is prepended to the sequence,

and visual frames and corresponding responses are interleaved. This design enables the model to learn temporal dependencies and identify appropriate response moments throughout the video stream.

For training data, we collect a diverse set of temporally annotated video datasets across multiple tasks, including dense video captioning [59; 60], sequential step recognition [61; 62], grounded video question answering [63; 64; 65], temporal video grounding [66], and temporal action detection [67; 68]. For each task, we design specific prompt templates and randomly select among them as $Q$ during training (details in Appendix A). To supervise activation timing, we insert the response $<A_i>$ at the end of its corresponding annotated timestamp (during inference, the response $<A_i>$ is generated by the larger main Video-LLM). Besides, only the last $P\%$ of frames of each video segment $V_i$ are labeled as positive (*i.e.,* response-worthy), while earlier frames are treated as negatives. $P$ is dynamically sampled between 0% and 50% for each training instance, simulating a variety of activation patterns and enhancing the model's robustness to temporal variations.

# 4 Stream-IT Dataset

As analyzed in Section 3.1, streaming scenarios are primarily characterized by multi-turn real-time understanding and proactive responses. However, existing datasets and video sources fall short of fully supporting these requirements [69; 70]. To fill this gap and further enhance the streaming interaction capability of *StreamBridge*, we introduce *Stream-IT*—a video-text dataset specifically designed for streaming instruction tuning with an interleaved multi-turn dialogue format.

**Datasets for Proactive Understanding.** We collect a set of public datasets enriched with timestamp annotations, spanning a wide range of tasks including: *(i) Dense Video Captioning* [59; 60; 71]; *(ii) Sequential Step Recognition* [61; 62; 72]; *(iii) Grounded VideoQA* [63; 73; 74; 75]. All datasets are reformatted into a proactive-style interleaved format: '$<Q><V_1><A_1>, <V_2><A_2>, \cdots$', where $Q$ may be an open-ended query (*e.g.,* "Who is the man going to find?") or a goal-oriented instruction (*e.g.,* "Show me all the steps for cooking."). Unlike traditional single-turn datasets where a question is immediately followed by an answer [69; 70], our structure introduces a temporal delay between $<Q>$ and $<A>$ through the inserted video segments $<V>$, simulating proactive response scenarios.

**StreamingQA-120K: Multi-Turn, Long-Form QA Construction.** To further support long-context, multi-turn real-time understanding, we introduce *StreamingQA-120K*, a large-scale synthetic dataset constructed by composing long-form videos from existing short video clips. Labeling long-duration videos with dense multi-turn QA pairs is prohibitively expensive. To address this, we leverage short clips from large-scale video-caption datasets, including WebVid-10M [19], Panda-70M [18], and InternVid-10M [17]. We filter approximately 1.28 million clips using semantic similarity between video and caption to ensure alignment, with each clip being around 12 seconds long. With these short clips, to form coherent long-form videos, we then iteratively compute pairwise similarity between videos and concatenate highly similar clips. Each constructed video contains roughly 10 clips, with an average length exceeding 150 seconds. Captions for each clip are preserved with natural timestamps. Using these captions, we employ GPT-4o [22] to generate diverse question-answer pairs spanning 8 task types. By default, each QA pair $<Q_i><A_i>$ is inserted immediately after its corresponding clip $<V_i>$, forming sequences like '$<V_1><Q_1><A_1>, <V_2><Q_2><A_2>, ...$'. Here, we introduce two augmentation strategies during sequence construction:

- **Random QA Drop**: randomly drops some QA pairs by transforming '$<V_i><Q_i><A_i>$' to '$<V_i>$' with a probability of $P_{\text{drop}}$, to prevent overfitting to fixed QA positions and enhance the model's robustness in temporal variations. We set $P_{\text{drop}}$ to be 0.55 by default.

- **QA Interval Shift**: with probability $P_{\text{shift}}$, transforms sequences from '$<V_i><Q_i><A_i>$' to '$<Q_i><V_i><A_i>$', where the visual content $V_i$ serves as the temporal delay between question and response for proactive scenarios. $P_{\text{shift}}$ is set to 0.1 here.

Together, these strategies ensure that the *Stream-IT* dataset supports rich and varied streaming interaction formats, enabling both multi-turn real-time dialogue and proactive response capabilities across a wide range of tasks and timescales. More details on data statistics, concatenation strategy of StreamingQA-120K, and prompts for QA generation are provided in Appendix B.

| Method | # of Frames | OVO-Bench Real-Time | | | | | | | Streaming-Bench Real-Time | | | | | | | | | | |
|---|---|---|---|---|---|---|---|---|---|---|---|---|---|---|---|---|---|---|---|
| | | OCR | ACR | ATR | STU | FPD | OJR | AVG. | OP | CR | CS | ATP | EU | TR | PR | SU | ACP | CT | AVG. |
| **Human** | | | | | | | | | | | | | | | | | | | |
| Human | - | 93.96 | 92.57 | 94.83 | 92.70 | 91.09 | 94.02 | 93.20 | 89.47 | 92.00 | 93.60 | 91.47 | 95.65 | 92.52 | 88.00 | 88.75 | 89.74 | 91.30 | 91.46 |
| **Proprietary Models (Offline), Single-Turn Evaluation** | | | | | | | | | | | | | | | | | | | |
| Gemini 1.5 pro [23] | 1 FPS | 85.91 | 66.97 | 79.31 | 58.43 | 63.37 | 61.96 | 69.32 | 79.02 | 80.47 | 83.54 | 79.67 | 80.00 | 84.74 | 77.78 | 64.23 | 71.95 | 48.70 | 75.69 |
| GPT-4o [22] | 64 | 69.80 | 64.22 | 71.55 | 51.12 | 70.3 | 59.78 | 64.46 | 77.11 | 80.47 | 83.91 | 76.47 | 70.19 | 83.80 | 66.67 | 62.19 | 69.12 | 49.22 | 73.28 |
| **Open-Source Models (Offline), Single-Turn Evaluation** | | | | | | | | | | | | | | | | | | | |
| Qwen2-VL-72B [2] | 64 | 65.77 | 60.55 | 69.83 | 51.69 | 69.31 | 54.35 | 61.92 | - | - | - | - | - | - | - | - | - | - | - |
| LLaVA-Video-7B [15] | 64 | 69.13 | 58.72 | 68.83 | 49.44 | 74.26 | 59.78 | 63.52 | - | - | - | - | - | - | - | - | - | - | - |
| LLaVA-OV-7B [3] | 64/32 | 66.44 | 57.80 | 73.28 | 53.37 | 71.29 | 61.96 | 64.02 | 80.38 | 74.22 | 76.03 | 80.72 | 72.67 | 71.65 | 67.59 | 65.45 | 65.72 | 45.08 | 71.12 |
| Qwen2-VL-7B [2] | 64/1 FPS | 60.40 | 50.46 | 56.03 | 47.19 | 66.34 | 55.43 | 55.98 | 75.20 | 82.81 | 73.19 | 77.45 | 68.32 | 71.03 | 72.22 | 61.19 | 61.47 | 46.11 | 69.04 |
| InternVL-V2-8B [76] | 64/16 | 67.11 | 60.55 | 63.79 | 46.07 | 68.32 | 56.52 | 60.39 | 68.12 | 60.94 | 69.40 | 77.12 | 67.70 | 62.93 | 59.26 | 53.25 | 54.96 | 56.48 | 63.72 |
| **Open-Source Models (Streaming), Single-Turn Evaluation** | | | | | | | | | | | | | | | | | | | |
| Flash-VStream-7B [11] | 1 FPS | 24.16 | 29.36 | 28.45 | 33.71 | 25.74 | 28.80 | 28.37 | 25.89 | 43.57 | 24.91 | 23.87 | 27.33 | 13.08 | 18.52 | 25.20 | 23.87 | 48.70 | 23.23 |
| VideoLLM-Online-8B [10] | 2 FPS | 8.05 | 23.85 | 12.07 | 14.04 | 45.54 | 21.20 | 20.79 | 39.07 | 40.06 | 34.49 | 31.05 | 45.96 | 32.40 | 31.48 | 34.16 | 42.49 | 27.89 | 35.99 |
| Dispider [13] | 1 FPS | 57.72 | 49.54 | 62.07 | 44.94 | 61.39 | 51.63 | 54.55 | 74.92 | 75.53 | 74.10 | 73.08 | 74.44 | 59.92 | 76.14 | 62.91 | 62.16 | 45.80 | 67.63 |
| **Models under *StreamBridge* (Offline → Streaming), Multi-Turn Evaluation** | | | | | | | | | | | | | | | | | | | |
| Oryx-1.5-7B† [1] | 1 FPS | 60.40 | 52.29 | 69.83 | 50.00 | 65.35 | 57.61 | 59.25 | 78.47 | 77.17 | 83.86 | 80.20 | 71.07 | 66.98 | 79.63 | 61.38 | 66.29 | 40.93 | 70.59 |
| *+ Stream-IT* | 1 FPS | 84.56 | 75.23 | 70.69 | 50.56 | 74.26 | 71.74 | 71.17 | 82.29 | 77.95 | 87.98 | 86.47 | 77.99 | 81.31 | 76.85 | 69.92 | 71.96 | 35.23 | 74.79 |
| LLaVA-OV-7B† [3] | 1 FPS | 58.39 | 59.63 | 69.82 | 44.38 | 76.23 | 61.41 | 61.64 | 76.84 | 77.17 | 82.60 | 75.25 | 64.15 | 64.17 | 75.00 | 61.38 | 61.19 | 46.11 | 68.39 |
| *+ Stream-IT* | 1 FPS | 74.50 | 77.06 | 70.69 | 54.49 | 73.27 | 69.57 | 69.93 | 82.29 | 72.44 | 92.09 | 80.86 | 71.07 | 74.46 | 75.00 | 62.20 | 70.26 | 28.50 | 70.92 |
| Qwen2-VL-7B† [2] | 1 FPS | 65.10 | 64.22 | 64.66 | 46.63 | 74.26 | 65.22 | 63.35 | 80.38 | 78.74 | 83.22 | 79.86 | 74.21 | 69.47 | 77.78 | 63.41 | 69.97 | 43.01 | 72.01 |
| *+ Stream-IT* | 1 FPS | 84.56 | 71.56 | 74.14 | 49.44 | 75.25 | 72.83 | **71.30** | 84.74 | 82.68 | 88.92 | 89.77 | 77.36 | 85.36 | 84.26 | 69.92 | 71.67 | 35.75 | **77.04** |

Table 1: Results on real-time understanding tasks on OVO-Bench and Streaming-Bench. † means models under *StreamBridge* framework, and + *Stream-IT* means finetuned on *Stream-IT*.

# 5 Experiments

## 5.1 Settings

**Models and Datasets.** We evaluate *StreamBridge* framework using three mainstream offline Video-LLMs to show its generalizability: LLaVA-OV-7B [3], Qwen2-VL-7B [2], and Oryx-1.5-7B [1]. To preserve their general video understanding capabilities during streaming adaptation, we supplement *Stream-IT* with approximately 600K samples from the LLaVA-178K [15], VCG-Plus [35] and ShareGPT4Video [16]. For the activation model, we fine-tune LLaVA-OV-0.5B [3] on our collected activation datasets as described in Sec. 3.2.3. The videos are sampled at 1 FPS. In Section 5.3, we use Qwen2-VL-7B as the default model unless otherwise specified. See the Appendix C for more details.

**Benchmarks.** For multi-turn real-time understanding, we choose OVO-Bench [20] and Streaming-Bench [21]. We primarily focus on their real-time tasks. For general video understanding, we evaluate our method across 7 video benchmarks, including 3 short-video benchmarks: MVBench [24], PerceptionTest [26], TempCompass [77], and 4 long-video benchmarks: EgoSchema [28], LongVideoBench [29], MLVU [27], and VideoMME [25]. To evaluate the proactive capability of our method, we use subtasks from ET-Bench [66] following previous works. See Appendix D for more benchmark details and evaluation metrics.

## 5.2 Main Results

**Multi-Turn Real-Time Understanding.** As discussed in Section 3.1, the results reported in the original paper [20; 21] in Table 1, marked as " (Offline), Single-Turn Evaluation ", do not reflect performance in real streaming scenarios. They segment a complete video into several individual clips, discarding historical visual and dialogue contexts, thereby limiting the upper bound of the performance. In contrast, with the *StreamBridge* framework, denoted as " (Offline → Streaming), Multi-Turn Evaluation ", these offline models are equipped to process streaming videos at 1 FPS in a multi-turn manner, while maintaining input length and historical contexts within a predefined maximum token budget.

Specifically, we observe that Qwen2-VL† demonstrates notable improvements in the streaming setting, with its average score on OVO-Bench increasing from 55.98 to 63.35, and on Streaming-Bench from 69.04 to 72.01. Conversely, LLaVA-OV† shows a slight performance drop when transitioning to the streaming setup: from 64.02 to 61.64 on OVO-Bench, and from 71.12 to 68.39 on Streaming-Bench. We attribute these differences to the nature of their pretraining data, where Qwen2-VL benefits from richer interleaved multimodal training (e.g., image/video-text sequences), which makes it more adept at understanding interleaved video-text inputs and utilizing extended context effectively. On

| Model | MVBench | PerceptionTest | TempCompass | EgoSchema | LongVideoBench | MLVU | VideoMME (w/o subs) |
|---|---|---|---|---|---|---|---|
| | Avg | Val | MC | Test | Val | M-Avg | Avg |
| Avg. Duration | 16s | 23s | 12s | 180s | 473s | 651s | 1010s |
| **Proprietary Models** | | | | | | | |
| Gemini 1.5 pro [23] | 60.5 | - | 67.1 | 71.2 | 64.0 | - | 75.0 |
| GPT-4o [22] | 64.6 | - | 70.9 | 72.2 | 66.7 | 64.6 | 71.9 |
| **Open-Source Models** | | | | | | | |
| Kangaroo-8B [78] | 61.0 | - | 62.5 | - | 54.8 | 61.0 | 56.0 |
| LongVILA-7B [79] | - | - | - | 67.7 | - | - | 57.5 |
| LongVU-7B [80] | 66.9 | - | - | 67.6 | - | 65.4 | 60.6 |
| Apollo-7B [4] | - | 67.3 | 64.9 | - | 58.5 | 70.9 | 61.3 |
| NVILA-8B [81] | 68.1 | 65.4 | 69.7 | - | 57.7 | 70.1 | 64.2 |
| SF-LLaVA-1.5-7B [5] | - | 69.6 | 68.8 | - | 62.5 | 71.5 | 63.9 |
| InternVL2.5-8B [82] | 72.0 | 68.2 | 68.3 | 51.5 | 60.0 | 68.9 | 64.2 |
| VideoChat-Flash-7B [83] | 74.0 | 76.2 | - | - | 64.7 | 74.7 | 65.3 |
| VideoLLaMA3-7B [37] | 69.7 | 72.8 | 68.1 | 63.3 | 59.8 | 73.0 | 66.2 |
| Oryx-1.5-7B [1] | 67.6 | 70.0 | 58.8 | - | 56.3 | 67.5 | 58.8 |
| Oryx-1.5-7B (ours) ‡ | 68.0 (↑0.4) | 71.0 (↑1.0) | 69.0 (↑10.2) | 61.2 | 58.9 (↑2.6) | 71.4 (↑4.0) | 65.5 (↑6.7) |
| LLaVA-OV-7B [3] | 56.7 | 57.1 | 64.8 | 60.1 | 56.3 | 64.7 | 58.2 |
| LLaVA-OV-7B (ours) ‡ | 59.4 (↑2.7) | 63.9 (↑6.8) | 67.7 (↑2.9) | 67.0 (↑6.9) | 54.3 (↓2.0) | 68.2 (↑3.5) | 61.2 (↑3.0) |
| Qwen2-VL-7B [2] | 67.0 | 62.3 | 67.9 | 66.7 | - | - | 63.3 |
| Qwen2-VL-7B (ours) ‡ | 64.4 (↓2.6) | 69.9 (↑7.6) | 71.1 (↑3.2) | 66.9 (↑0.2) | 59.1 | 69.6 | 64.4 (↑1.1) |

Table 2: Results on general video understanding benchmarks. ‡ means models under *Stream-Bridge* framework and fine-tuned on *Stream-IT*.

the other hand, LLaVA-OV is trained with fewer interleaved sequences, making it less suited for multi-turn streaming inputs. When faced with long, interleaved video-text sequences in streaming scenarios, its performance tends to degrade as more historical frames accumulate. Notably, fine-tuning these models on the proposed *Stream-IT* leads to substantial improvements in multi-turn real-time understanding. For instance, Oryx-1.5[†] achieves a performance gain of +11.92 on OVO-Bench and +4.2 on Streaming-Bench. Furthermore, Qwen2-VL[†] reaches an average score of 71.30 on OVO-Bench and 77.04 on Streaming-Bench, outperforming proprietary models such as GPT-4o and Gemini 1.5 Pro. These results validate the effectiveness of both our *StreamBridge* framework and the *Stream-IT* dataset in enhancing multi-turn real-time understanding in streaming scenarios.

**General Video Understanding.** While our method is designed for online scenarios, we also verify that it does not downgrade the base model's performance on standard offline video tasks. As shown in Table 2, models equipped with the *StreamBridge* framework and fine-tuned on *Stream-IT* (denoted with ‡) exhibit consistent improvements or maintain comparable performance relative to their original versions. For instance, Oryx-1.5-7B[‡] achieves 65.5 on the challenging VideoMME with an increase of 6.7, while LLaVA-OV-7B[‡] outperforms its base model across nearly all benchmarks, except LongVideoBench. Likewise, Qwen2-VL-7B[‡] achieves competitive results on MVBench, while surpassing its original counterpart on other benchmarks. These outcomes demonstrate that our streaming adaptation enables models to retain, or even exceed their original performance in general video understanding tasks, demonstrating the generality and non-degradability of our method.

**Online Activation.** We evaluate the proactive capability of our framework in Table 3. Notably, in all tasks, the question is presented at the beginning of the video, and the model must autonomously decide when to respond. On the ET-Bench, *Stream-Bridge* outperforms both VideoLLM-Online [10] and Dispider [13] across generation-based tasks such as DVC (Dense Video Captioning) and SLC (Step Localization and Captioning),

| Method | # of | ET-Bench | | | | | |
|---|---|---|---|---|---|---|---|
| | Frames | $TVG_{F1}$ | $TAL_{F1}$ | $DVC_{F1}$ | $DVC_{Sim}$ | $SLC_{F1}$ | $SLC_{Sim}$ |
| VideoLLM-Online [10] | 2 FPS | 13.2 | 9.1 | 24.0 | 13.4 | 9.9 | 10.1 |
| Dispider [13] | 1 FPS | **36.1** | **27.3** | 33.8 | 18.9 | 18.8 | 12.4 |
| **Models under *StreamBridge* Framework** | | | | | | | |
| Oryx-1.5 (ours)‡ | 1 FPS | 34.3 | 24.3 | 37.8 | 24.0 | 22.5 | **17.3** |
| LLaVA-OV (ours)‡ | 1 FPS | 34.3 | 24.3 | 37.9 | 24.2 | **22.8** | 16.2 |
| Qwen2-VL (ours)‡ | 1 FPS | 34.3 | 24.3 | **38.3** | **25.1** | 22.6 | 17.1 |

Table 3: Results on ET-Bench. ‡ denotes models under *StreamBridge* framework and fine-tuned on *Stream-IT*. $TVG_{F1}$ and $TAL_{F1}$ scores are identical across *Stream-Bridge* models due to sharing the same activation model.

achieving higher similarity scores of $DVC_{Sim}$ and $SLC_{Sim}$, by producing more accurate and context-aware descriptions in streaming scenarios. We attribute this to the decoupled nature of the activation model, which enables the main Video-LLM to focus solely on video understanding and language generation, free from the burden of proactive decision-making. We also observe that Qwen2-VL[‡] achieves better text similarity scores than other Video-LLMs, consistent with its strong real-time understanding performance presented in Table 1.

## 5.3 In-Depth Analysis

| Compression | OVO | Streaming | ET | |
|---|---|---|---|---|
| | Avg. | Avg. | $DVC_{Sim}$ | $SLC_{Sim}$ |
| Truncation | 68.88 | 72.79 | 22.1 | 16.7 |
| Round-Uniform | 69.91 | 74.18 | 23.8 | 15.9 |
| Round-Decayed | **71.30** | **77.04** | **25.1** | **17.1** |

Table 4: Ablation studies on different compression strategies.

| LLaVA-178k | Stream-IT | | OVO | Streaming | MVBench | VideoMME |
|---|---|---|---|---|---|---|
| (600k used) | w/o SQA-120k | w/ SQA-120k | Avg. | Avg. | Avg. | Overall. |
| ✓ | | | 65.98 | 71.36 | **64.5** | 61.7 |
| | | ✓ | 71.28 | 74.10 | 58.8 | 59.0 |
| ✓ | ✓ | | 67.67 | 72.42 | 63.1 | 63.6 |
| ✓ | | ✓ | **71.30** | **77.04** | 64.4 | **64.4** |

Table 5: Ablation studies on *Stream-IT*. SQA-120k denotes the generated StreamingQA-120k.

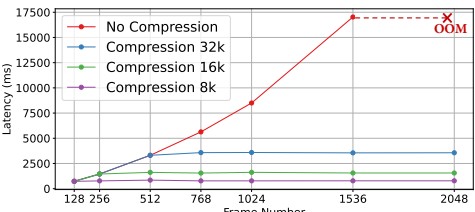

Figure 4: Inference Latency (y-axis) vs. Frame Number (x-axis).

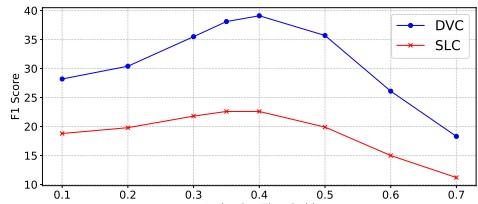

Figure 5: Ablation studies on the activation threshold $\alpha$.

**Round-Decayed Compression.** We set the maximum input length $\mathrm{MaxLen} = 16384$, and denote the current length of the input embeddings as $L$. To assess the effectiveness of our round-decayed compression strategy, we compare it against two alternative methods: (1) Truncation: If $L > \mathrm{MaxLen}$, only keep the last $L$ tokens in the input sequence. (2) Round-Uniform: We treat each round equally by reducing the number of visual tokens with a fixed ratio $\frac{L-\mathrm{MaxLen}}{L}$ per round, to keep the total length within $\mathrm{MaxLen}$. The results are reported in Table 4. We observe that Truncation yields the worst performance, as it indiscriminately removes both visual and textual history tokens, severely weakening multi-turn reasoning. The Round-Uniform strategy performs slightly better, but still underperforms our method. It compresses the latest visual tokens, which are critical for real-time comprehension, thus leading to degraded performance, particularly on OVO-Bench and Streaming-Bench.

**Inference Latency.** We also evaluate the inference latency on a single A100-80G GPU with different $\mathrm{MaxLen}$ (8k, 16k, 32k), as shown in Figure 4. Our results show that our compression method maintains near-constant latency when the number of input tokens exceeds $\mathrm{MaxLen}$, whereas models without compression suffer from sharply increasing delays and eventually trigger out-of-memory (OOM) errors with 2048 frames. This highlights the necessity of effective compression to balance inference efficiency and memory usage in streaming settings.

**Impact of *Stream-IT*.** Table 5 ablates effectiveness of *Stream-IT*. Training on LLaVA-178K alone causes a marked drop on both OVO-Bench and Streaming-Bench, as it lacks interleaved video–text samples necessary for multi-turn interactions. Conversely, using only *Stream-IT* without LLaVA-178K leads to declines in general video understanding (MVBench, VideoMME), indicating that the larger offline data corpus still contributes valuable world knowledge. Finally, removing the synthetic StreamingQA-120K subset from *Stream-IT* degrades performance across both streaming and offline benchmarks, underscoring the crucial role of StreamingQA-120K in boosting both streaming and offline video understanding capabilities.

**Impact of $\mathrm{MaxLen}$.** To better understand the impact of $\mathrm{MaxLen}$, we conducted ablation studies using the Qwen2-VL-StreamBridge model with 1 FPS sampling, varying $\mathrm{MaxLen}$ from 4k to 32k. From Table 6, we observe the following: (1) For streaming tasks (e.g., OVO-Bench Real-Time): Model performance remains relatively stable across varying $\mathrm{MaxLen}$ values, ranging from 70.49% to 71.30%; Accuracy peaks at 16k and slightly declines at 32k, suggesting that further increasing the memory budget yields

| $\mathrm{MaxLen}$ | OVO-Bench | VideoMME |
|---|---|---|
| | (Real-Time) Avg. | Avg. |
| 4k | 70.49 | 61.7 |
| 8k | 70.89 | 63.6 |
| 16k | **71.30** | 64.4 |
| 32k | 71.16 | **64.7** |

Table 6: Ablation studies on $\mathrm{MaxLen}$

diminishing returns. This supports our design assumption: in streaming scenarios, models primarily rely on recent context, and older frames can be compressed without significant performance loss. (2) For offline tasks (e.g., VideoMME): Accuracy improves consistently as $\mathrm{MaxLen}$ increases, from

61.7% at 4k to 64.7% at 32k. This means that offline tasks benefit more from retaining the full temporal context and more uncompressed video tokens, especially for long videos that require detailed long-range understanding. *StreamBridge* can flexibly balance efficiency and performance across both streaming and offline settings by adjusting the memory budget accordingly, and we set $\mathrm{MaxLen} = 16k$ to strike a good balance between them across most tasks.

**Activation Threshold.** The compact activation model makes a per-frame decision to trigger responses, with frequency determined by the activation threshold $\alpha$ (see score head in Figure 3). We adopt a default $\alpha$ of 0.35, following common practice [52; 14]. Figure 5 illustrates the impact of varying this threshold: both excessively low and high values of $\alpha$ decrease F1 scores ($DVC_{F1}$ and $SLC_{F1}$ on ET-Bench). A low threshold triggers overly frequent responses, while a high threshold suppresses them excessively, both of which hurt performance. Nonetheless, this hyper-parameter allows users to flexibly control response frequency through $\alpha$, adapting to different practical scenarios.

## 6 Conclusion

We present *StreamBridge*, a novel framework that transforms offline Video-LLMs into streaming-capable models. *StreamBridge* introduces a memory buffer paired with a round-decayed compression strategy, and decouples the activation function with a compact activation model. We also construct *Stream-IT*, a dataset with interleaved video-text sequences to further support *StreamBridge*. Extensive experiments on diverse benchmarks demonstrate that our method not only preserves the strengths of the base models but also equips them with the ability to make timely, proactive responses across multi-turn, long-context streaming scenarios. We believe *StreamBridge* offers a general solution for bridging the gap between offline Video-LLMs and real-world, interactive streaming applications.

**Acknowledgment.** We thank Yu Liu, Haiming Gang, and Mingfei Gao for their kind help.

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

Table 7: Prompts used for datasets to train the activation model.

## A Datasets Used to Train the Activation Model

To train the activation model $\mathcal{ACT}(\cdot)$, we compile a diverse collection of video datasets spanning five distinct tasks:

- **Dense Video Captioning:** ActivityNet Captions [59], Shot2Story [60].

- **Sequential Step Recognition:** YouCook2 [62], COIN [61].

- **Temporal Action Detection:** FineAction [67], HACS [68].

- **Grounded VideoQA:** Multihop-EgoQA [64], EgoTimeQA [73].

- **Temporal Video Grounding:** Charades [84], and the TVG subset from ET-Instruct [66].

In total, our training set contains approximately 180k video samples. For each sample, we construct an input prompt using task-specific templates. A prompt is randomly sampled from a predefined pool for the corresponding task to ensure stylistic diversity and improve generalization across video domains. The full list of prompt templates is provided in Table 7. During training, the prompt is inserted at the beginning of the input sequence as in Figure 3.

Understand the following different task descriptions:
<task descriptions>

**USER**:
Now, please carefully review the following video caption:
<caption>
According to the given caption, please select ONE task type that is best suitable to generate the QA pair, and output your question, answer and task type following the SAME FORMAT as the examples above. Remember, just generate only ONE QA pair.

Table 8: Prompts used to generate QA pairs with GPT-4o.

# B  *Stream-IT* Construction

## B.1  Statistics of *Stream-IT*

We provide detailed statistics of the *Stream-IT* dataset in Table 9, including the number of samples, average video duration, and the corresponding source datasets used for each task. Notably, during the construction of the dense video captioning tasks, including ActivityNet[59] and Shot2Story [60], we only arrange 20% of the sequences with the proactive format of '$<Q> <V_1> <A_1>, <V_2> <A_2>, \cdots$', while the 80% of the sequences with the multi-turn format of '$<V_1> <Q_1> <A_1>, <V_2> <Q_2> <A_2>$, ...', where $<Q_i>$ is the question asking about current situations like 'What is happening now?'.

## B.2  Concatenation Strategy for Constructing StreamingQA-120K

Starting from a pool of 1.28 million filtered short videos sourced from WebVid-10M [19], Panda-70M [18], and InternVid-10M [17], our goal is to iteratively merge semantically similar clips to form long-form video samples. Let $\mathcal{V}$ denote the entire set of filtered videos. We initiate the process by randomly sampling one clip $\mathcal{V}_1$ from $\mathcal{V}$ as the anchor. We then compute pairwise semantic similarity (based on the middle frame) between $\mathcal{V}_1$ and all other videos in $\mathcal{V} \setminus \mathcal{V}_1$. A new clip $\mathcal{V}_2$ is sampled according to the similarity distribution (without replacement). The procedure is repeated using $\mathcal{V}_2$ as the new anchor, generating $\mathcal{V}_3$ from $\mathcal{V} \setminus \mathcal{V}_1, \mathcal{V}_2$, and so on. This results in a similarity-ordered list of videos $\mathcal{V}_1, \mathcal{V}_2, \mathcal{V}_3, \ldots$. We formulate this process in Algorithm 2. This approach allows for flexible concatenation of any number $k$ of clips to construct a long-form sample, by directly selecting

| Task | # of Samples | Datasets | Average duration |
|------|--------------|----------|------------------|
| Dense Video Captioning | ~54k | ActivityNet [59] (~10k)
Shot2Story [60] (~36k)
ViTT [71] (~8k) | ~180s
~16s
~210s |
| Sequential Step Recognition | ~22k | YouCook2 [62] (~1.3k)
COIN [61] (~11k)
HowToStep [72] (~10k) | ~317s
~145s
~190s |
| Grounded Video Question Answering | ~69k | MovieChat [74] (~0.8k)
EgoTimeQA [73] (~10k)
QAEgo4D [63] (~15k)
FineVideo [75] (~43k) | ~10k frames
~150s
~495s
~280s |
| Multi-turn Real-time Question Answering | ~120k | StreamingQA-120K (~120k)
(Sourced from Webvid-10M[19], Panda-70M[18], InternVid-10M[17]) | ~150s |

Table 9: Involved tasks and datasets in *Stream-IT*.

---

**Algorithm 2:** Constructing Similarity-Ordered Video Clip Sequence

1 **Inputs:** Pool of filtered short video clips $\mathcal{V}_{\text{pool}} = \{v_1, v_2, \ldots, v_M\}$;
2 **Initializations:** $\mathcal{V}_{\text{ordered}} = [\quad]$, $v_{\text{anchor}} = \text{None}$;
3 **Define:** $Sim(v_{\text{anchor}}, \mathcal{C}_{\text{candidates}})$: function that returns the clip from $\mathcal{C}_{\text{candidates}}$ most similar to $v_{\text{anchor}}$.
4 $v_{\text{anchor}} \leftarrow \text{RandomSample}(\mathcal{V}_{\text{pool}})$ ;                    // Randomly select the first anchor clip
5 $\mathcal{V}_{\text{ordered}} \leftarrow v_{\text{anchor}}$ ;                                          // Add $v_{\text{anchor}}$ to $\mathcal{V}_{\text{ordered}}$
6 $\mathcal{V}_{\text{pool}} \leftarrow \mathcal{V}_{\text{pool}} \setminus \{v_{\text{anchor}}\}$ ;                         // Remove $v_{\text{anchor}}$ from $\mathcal{V}_{\text{pool}}$
7 **while** $\mathcal{V}_{\text{pool}}$ is not empty **do**
8      $v_{\text{next}} \leftarrow Sim(v_{\text{anchor}}, \mathcal{V}_{\text{pool}})$ ;                 // Find the clip in pool most similar to $v_{\text{anchor}}$
9      $\mathcal{V}_{\text{ordered}} \leftarrow v_{\text{next}}$
10      $\mathcal{V}_{\text{pool}} \leftarrow \mathcal{V}_{\text{pool}} \setminus \{v_{\text{next}}\}$
11      $v_{\text{anchor}} \leftarrow v_{\text{next}}$ ;                                     // Update anchor to the newly added clip
12 **Output:** Similarity-ordered list of video clips $\mathcal{V}_{ordered} = [\mathcal{V}_1, \mathcal{V}_2, \ldots, \mathcal{V}_M]$.

---

a continuous span $\mathcal{V}_{[i:i+k]}$, without re-computing similarity each time. We also prepare hallucination questions irrelevant to existing video inputs following [20] with a ratio of 0.01%.

### B.3 Prompt Templates for Generating QA Pairs

To generate question-answer pairs based on clip-level captions, we design diverse prompt templates for 8 distinct reasoning tasks. Table 8 provides examples of these templates. Below, we summarize the <task descriptions> associated with each QA category:

- **[OP] Object Perception**: Detect and identify objects, focusing on recognizing their attributes in real time.

- **[AR] Action Recognition**: Identify human actions and interactions occurring in the current moment.

- **[SA] Spatial Awareness**: Understand spatial relationships among objects and events; reason about location, orientation, and distance.

- **[SR] Sequential Relationship**: Identify the temporal order of events and actions, especially those involving "before" and "after" cues.

- **[CR] Causal Reasoning**: Analyze cause-and-effect relationships between actions and outcomes.

- **[OCR] Optical Character Recognition**: Recognize and interpret textual content in scenes (e.g., subtitles, signs, charts).

- **[UEH] Unexpected Event Handling**: Detect and react to anomalies or sudden changes in the environment.

- **[EU] Event Understanding**: Summarize and reason about sequences of temporally linked events.

These diverse prompts ensure broad task coverage and help enhance the model's generalization across different temporal and semantic understanding challenges.

## C  More Implementation Details

For the main VideoLLMs, we use the following configurations for each model:

- LLaVA-OV-7B: We apply center cropping with a resolution of 384×384 and use a ×4 down sampler (bilinear interpolation) with the frame features, resulting in 49 tokens per frame.
- Oryx-1.5-7B: We use the model's default dynamic resolution, ranging from 288 to 480 pixels. With a ×4 down sampler on the frame features (average pooling), the resulting token count per frame varies between 33 and 59.
- Qwen2-VL-7B: The model uses a dynamic resolution between 224 and 448, with ×4 down sampling (average pooling) on the frame features, resulting in 36–64 tokens per frame.

All models are fine-tuned for one epoch using a learning rate of 2e-5 with a cosine annealing scheduler and AdamW optimizer. The image encoder remains frozen, while the visual projector and the LLM are fully trainable. The maximum length $MaxLen$ of input embeddings is set to 16384 for the round-decayed compression.

For the activation model, we adopt LLaVA-OV-0.5B as the base model. To improve efficiency, we aggressively pool the frame representations to 16 tokens per frame. During training, only the LoRA adapters, the projector, the score head, and the learnable activation token are trainable. The model is trained for 5 epochs using a fixed learning rate of 2e-5 for the projector, while 2e-4 for the LoRA adapters, score head, and the learnable activation token, with AdamW optimizer.

Notably, for both the main VideoLLM and the activation model, we sample frames at 1 FPS to better simulate real-world frame rates. For videos longer than 256 seconds, we uniformly sample 256 frames to fit within the maximum input length constraint. Experiments are conducted on NVIDIA-H100/A100 GPUs. During inference, we sample videos at 2 FPS for short video benchmarks like MVBench, PerpecptionTest, and TempCompass, while 1 FPS for multi-turn real-time understanding benchmarks and long video benchmarks including OVO-Bench, Streaming-Bench, MLVU, LongVideoBench, VideoMME, and EgoSchema.

## D  Benchmarks and Metrics

**Multi-turn Real-time Understanding Benchmarks.** We evaluate our method on two recently proposed large-scale streaming video benchmarks: OVO-Bench [20] and Streaming-Bench [21]. Both benchmarks are designed to assess streaming video comprehension under long-context, multi-turn settings. Our evaluation primarily focuses on their real-time understanding tasks. OVO-Bench contains 512 videos with an average length of 435 seconds and approximately 1,600 questions. The evaluated tasks include: (1) Spatial Understanding (STU), (2) Object Recognition (OJR), (3) Attribute Recognition (ATR), (4) Action Recognition (ACR), (5) Optical Character Recognition (OCR), and (6) Future Prediction (FPD). Streaming-Bench consists of 500 videos with an average length of 606 seconds and approximately 2,500 questions. It includes the following tasks: (1) Object Perception (OP), (2) Causal Reasoning (CR), (3) Clip Summarization (CS), (4) Attribute Perception (ATP), (5) Event Understanding (EU), (6) Text-Rich Understanding (TR), (7) Prospective Reasoning (PR), (8) Spatial Understanding (SU), (9) Action Perception (ACP), (10) Counting (CT). Both benchmarks are structured as multiple-choice question-answering tasks, and we report the accuracy.

**General Video Understanding Benchmarks.** To evaluate general video comprehension ability, we test our models on seven widely used benchmarks. This includes three short-video benchmarks: MVBench [24], Perception Test [26], and TempCompass [77], and four long-video benchmarks: EgoSchema [28], LongVideoBench [29], MLVU [27], and VideoMME [25]. These datasets span a broad range of video durations, from a few minutes to several hours. All are evaluated in a multiple-choice format, and accuracy is reported.

**Online Activation Benchmarks.** To assess the proactive capabilities of our framework, we evaluate performance on a subset of ET-Bench [66], including Temporal Video Grounding (TVG), Temporal Action Localization (TAL), Dense Video Captioning (DVC), and Sequential Localization Captioning (SLC). These tasks emphasize a shift from passive to active perception, requiring the model to determine when to respond based on upcoming visual inputs, rather than reacting immediately. For example, the Sequential Localization Captioning (SLC) task requires the model to both determine

the precise timing of a certain step and output its content. For evaluation metrics, we compute the average F1 score across multiple IoU thresholds (IoU $\in \{0.1, 0.3, 0.5, 0.7\}$) for localization-based tasks. For tasks involving text generation, we adopt sentence-level similarity metrics [85] to measure the semantic alignment between model outputs and ground-truth responses, following prior works [66; 13]. Specifically, the all-MiniLM-L6-v2 model in Sentence-Transformers library is used as the embedding model. Notably, in all these tasks, the question is presented at the beginning of the video, and the model must autonomously decide when to respond. Moreover, the results of $\text{TVG}_{F1}$, $\text{TAL}_{F1}$, are the same for our method with different main Video-LLMs, since they use the same activation model and will not be affected by the generated response.

# E    Broader Impacts

There are many real-world applications of streaming Video-LLMs, such as patient or elderly health monitoring, autonomous driving, and collaborative robots. However, there could be unintended usages and we advocate responsible usage complying with applicable laws and regulations.

# F    More Related Works

To address the challenge of long-context understanding in streaming video, several memory and retrieval mechanisms have been proposed. For instance, ReKV [56] introduces a training-free framework that stores and retrieves the Key-Value (KV) caches of processed frames, enabling offline models to answer user queries efficiently by reloading only the most relevant context. Besides, VideoStreaming [57] employs a memory-propagated encoding architecture where a condensed representation of the preceding clip serves as historical context for encoding the next, combined with an adaptive selection of memories for question-answering. Moreover, StreamChat [53] proposes a hierarchical memory system comprising short-term, long-term, and dialogue components to facilitate complex streaming interactions, and also contributes the StreamBench benchmark for evaluating diverse streaming scenarios. While these methods effectively advance long-context retention for reactive question-answering, *StreamBridge* differs by introducing a round-decayed compression strategy specifically tailored for multi-turn real-time interactions, which efficiently prunes redundant historical tokens while preserving recent context with high fidelity. Moreover, *StreamBridge* introduces a decoupled, lightweight activation model. This plug-and-play component operates in parallel with the main Video-LLM, enabling continuous proactive responses. These designs, supported by our dedicated Stream-IT dataset, effectively transform general-purpose offline models into versatile and proactive streaming assistants without compromising their core performance.

# G    Limitations

Although our proposed framework and dataset significantly enhance the streaming capabilities of existing offline Video-LLMs, there are still limitations worth noting. First, while *Stream-IT* provides large-scale multi-turn, interleaved training data, its construction relies partially on synthetic QA generation and clip concatenation, which, despite careful filtering, may introduce domain shift compared to truly continuous, real-world video streams. Future work could benefit from curating more organically collected long-form streaming videos with naturally evolving events and dialogues. Second, *StreamBridge* currently focuses on frame-by-frame streaming under relatively low sampling rates (e.g., 1 FPS). Extending the framework to handle denser frame rates or multi-modal streaming inputs (e.g., audio-visual-text) in real-time remains an important direction for future research.

## H Full Results on OVO-Bench and Streaming-Bench

| Method | # of Frames | Real-Time Visual Perception | | | | | | | Backward Tracing | | | | Forward Active Responding | | | | Overall. |
|---|---|---|---|---|---|---|---|---|---|---|---|---|---|---|---|---|---|
| | | OCR | ACR | ATR | STU | FPD | OJR | AVG. | EPM | ASI | HLD | AVG. | REC | SSR | CRR | AVG. | Overall AVG. |
| **Human** | | | | | | | | | | | | | | | | | |
| Human | - | 93.96 | 92.57 | 94.83 | 92.70 | 91.09 | 94.02 | 93.20 | 92.59 | 93.02 | 91.37 | 92.33 | 95.48 | 89.67 | 93.56 | 92.90 | 92.81 |
| **Proprietary Models (Offline), Single-Turn Evaluation** | | | | | | | | | | | | | | | | | |
| Gemini 1.5 pro [23] | 1 FPS | 85.91 | 66.97 | 79.31 | 58.43 | 63.37 | 61.96 | 69.32 | 58.59 | 76.35 | 52.64 | 62.54 | 35.53 | 74.24 | 61.67 | 57.15 | 63.00 |
| GPT-4o [22] | 64 | 69.80 | 64.22 | 71.55 | 51.12 | 70.30 | 59.78 | 64.46 | 57.91 | 75.68 | 48.66 | 60.75 | 27.58 | 73.21 | 59.40 | 53.40 | 59.54 |
| **Open-Source Models (Offline), Single-Turn Evaluation** | | | | | | | | | | | | | | | | | |
| Qwen2-VL-72B [2] | 64 | 65.77 | 60.55 | 69.83 | 51.69 | 69.31 | 54.35 | 61.92 | 52.53 | 60.81 | 57.53 | 56.95 | 38.83 | 64.07 | 45.00 | 49.30 | 56.27 |
| LLaVA-Video-7B [15] | 64 | 69.13 | 58.72 | 68.83 | 49.44 | 74.26 | 59.78 | 63.52 | 56.23 | 57.43 | 7.53 | 40.4 | 34.10 | 69.95 | 60.42 | 54.82 | 52.91 |
| LLaVA-OV-7B [3] | 64 | 66.44 | 57.80 | 73.28 | 53.37 | 71.29 | 61.96 | 64.02 | 54.21 | 55.41 | 21.51 | 43.71 | 25.64 | 67.09 | 58.75 | 50.50 | 52.74 |
| Qwen2-VL-7B [2] | 64 | 60.40 | 50.46 | 56.03 | 47.19 | 66.34 | 55.43 | 55.98 | 47.81 | 35.48 | 56.08 | 46.46 | 31.66 | 65.82 | 48.75 | 48.74 | 50.39 |
| InternVL-V2-8B [76] | 64 | 67.11 | 60.55 | 63.79 | 46.07 | 68.32 | 56.52 | 60.39 | 48.15 | 57.43 | 24.73 | 43.44 | 26.5 | 59.14 | 54.14 | 46.60 | 50.15 |
| **Open-Source Models (Streaming), Single-Turn Evaluation** | | | | | | | | | | | | | | | | | |
| Flash-VStream-7B [11] | 1 FPS | 24.16 | 29.36 | 28.45 | 33.71 | 25.74 | 28.80 | 28.37 | 39.06 | 37.16 | 5.91 | 27.38 | 8.02 | 67.25 | 60.00 | 45.09 | 33.61 |
| VideoLLM-Online-8B [10] | 2 FPS | 8.05 | 23.85 | 12.07 | 14.04 | 45.54 | 21.20 | 20.79 | 22.22 | 18.80 | 12.18 | 17.73 | - | - | - | - | - |
| Dispider [13] | 1 FPS | 57.72 | 49.54 | 62.07 | 44.94 | 61.39 | 51.63 | 54.55 | 48.48 | 55.41 | 4.30 | 36.06 | 18.05 | 37.36 | 48.75 | 34.72 | 41.78 |
| **Models under *StreamBridge* (Offline → Streaming), Multi-Turn Evaluation** | | | | | | | | | | | | | | | | | |
| Oryx-1.5-7B[†] [1] | 1 FPS | 60.40 | 52.29 | 69.83 | 50.00 | 65.35 | 57.61 | 59.25 | 54.21 | 55.41 | 5.40 | 38.33 | 20.65 | 37.56 | 40.00 | 32.74 | 43.44 |
| + *Stream-IT* | 1 FPS | 84.56 | 75.23 | 70.69 | 50.56 | 74.26 | 71.74 | 71.17 | 69.02 | 59.50 | 79.03 | 69.17 | 20.51 | 66.89 | 60.41 | 49.27 | 63.21 |
| LLaVA-OV-7B[†] [3] | 1 FPS | 58.39 | 59.63 | 69.82 | 44.38 | 76.23 | 61.41 | 61.64 | 53.87 | 54.72 | 30.64 | 46.41 | 14.41 | 51.23 | 43.33 | 36.33 | 48.13 |
| + *Stream-IT* | 1 FPS | 74.50 | 77.06 | 70.69 | 54.49 | 73.27 | 69.57 | 69.93 | 66.67 | 6149 | 85.48 | 71.21 | 17.83 | 66.06 | 61.67 | 48.52 | 63.22 |
| Qwen2-VL-7B[†] [2] | 1 FPS | 65.10 | 64.22 | 64.66 | 46.63 | 74.26 | 65.22 | 63.35 | 55.56 | 60.14 | 62.90 | 59.53 | 22.14 | 61.12 | 49.58 | 44.28 | 55.72 |
| + *Stream-IT* | 1 FPS | 84.56 | 71.56 | 74.14 | 49.44 | 75.25 | 72.83 | 71.30 | 67.68 | 57.43 | 79.03 | 68.05 | 19.17 | 64.25 | 61.67 | 48.36 | 62.57 |

Table 10: Full results on OVO-Bench. [†] means models under *StreamBridge* framework

| Method | # of Frames | Real-Time Visual Understanding | | | | | | | | | | Omni-Source Understanding | | | | | Contextual Understanding | | | | | Overall. |
|---|---|---|---|---|---|---|---|---|---|---|---|---|---|---|---|---|---|---|---|---|---|---|
| | | OP | CR | CS | ATP | EU | TR | PR | SU | ACP | CT | AVG. | ER | SCU | SD | MA | AVG. | ACU | MCU | SQA | PO | AVG. | Overall AVG. |
| **Human** | | | | | | | | | | | | | | | | | | | | | | | |
| Human | - | 89.47 | 92.00 | 93.60 | 91.47 | 95.65 | 92.52 | 88.00 | 88.75 | 89.74 | 91.30 | 91.46 | 88.00 | 88.24 | 93.60 | 90.27 | 90.26 | 88.80 | 90.40 | 95.00 | 100 | 93.55 | 91.66 |
| **Proprietary Models (Offline), Single-Turn Evaluation** | | | | | | | | | | | | | | | | | | | | | | | |
| Gemini 1.5 pro [23] | 1 FPS | 79.02 | 80.47 | 83.54 | 79.67 | 80.00 | 84.74 | 77.78 | 64.23 | 71.95 | 48.70 | 75.69 | 46.80 | 39.60 | 74.90 | 80.00 | 60.22 | 51.41 | 40.73 | 54.80 | 45.10 | 48.73 | 67.07 |
| GPT-4o [22] | 64 | 77.11 | 80.47 | 83.91 | 76.47 | 70.19 | 83.80 | 66.67 | 62.19 | 69.12 | 49.22 | 73.28 | 41.20 | 37.20 | 43.60 | 56.00 | 44.50 | 41.20 | 38.40 | 32.80 | 56.86 | 38.70 | 60.15 |
| **Open-Source Models (Offline), Single-Turn Evaluation** | | | | | | | | | | | | | | | | | | | | | | | |
| LLaVA-OV-7B [3] | 32 | 80.38 | 74.22 | 76.03 | 80.72 | 72.67 | 71.65 | 67.59 | 65.45 | 65.72 | 45.08 | 71.12 | 40.80 | 37.20 | 33.60 | 44.80 | 38.40 | 35.60 | 36.00 | 27.27 | 29.55 | 32.74 | 56.36 |
| Qwen2-VL-7B [2] | 0.2-1 FPS | 75.20 | 82.81 | 73.19 | 77.45 | 68.32 | 71.03 | 72.22 | 61.19 | 61.47 | 46.11 | 69.04 | 41.20 | 22.00 | 32.80 | 43.60 | 34.90 | 31.20 | 26.00 | 39.60 | 22.73 | 31.66 | 54.14 |
| InternVL-V2-8B [76] | 16 | 68.12 | 60.94 | 69.40 | 77.12 | 67.70 | 62.93 | 59.26 | 53.25 | 54.96 | 56.48 | 63.72 | 37.60 | 26.40 | 37.20 | 42.00 | 35.80 | 32.00 | 31.20 | 32.32 | 40.91 | 32.42 | 51.40 |
| **Open-Source Models (Streaming), Single-Turn Evaluation** | | | | | | | | | | | | | | | | | | | | | | | |
| Flash-VStream-7B [11] | 1 FPS | 25.89 | 43.57 | 24.91 | 23.87 | 27.33 | 13.08 | 18.52 | 25.20 | 23.87 | 48.70 | 23.23 | 25.91 | 24.90 | 25.60 | 28.40 | 26.00 | 24.80 | 25.20 | 26.80 | 1.96 | 24.12 | 24.04 |
| VideoLLM-Online-8B [10] | 2 FPS | 39.07 | 40.06 | 34.49 | 31.05 | 45.96 | 32.40 | 31.48 | 34.16 | 42.49 | 27.89 | 35.99 | 31.20 | 26.51 | 24.10 | 32.00 | 28.45 | 24.19 | 29.20 | 30.80 | 3.92 | 26.55 | 32.48 |
| Dispider [13] | 1 FPS | 74.92 | 75.53 | 74.10 | 73.08 | 74.44 | 59.92 | 76.14 | 62.91 | 62.16 | 45.80 | 67.63 | 35.46 | 25.26 | 38.57 | 43.34 | 35.66 | 39.62 | 27.65 | 34.80 | 25.34 | 33.61 | 53.12 |
| **Models under *StreamBridge* (Offline → Streaming), Multi-Turn Evaluation** | | | | | | | | | | | | | | | | | | | | | | | |
| Oryx-1.5-7B[†] [1] | 1 FPS | 78.47 | 77.17 | 83.86 | 80.20 | 71.07 | 66.98 | 79.63 | 61.38 | 66.29 | 40.93 | 70.59 | 30.00 | 15.20 | 33.60 | 43.20 | 30.50 | 20.40 | 24.80 | 39.60 | 54.90 | 34.93 | 53.76 |
| + *Stream-IT* | 1 FPS | 82.29 | 77.95 | 87.98 | 86.47 | 77.99 | 81.31 | 76.85 | 69.92 | 71.96 | 35.23 | 74.79 | 19.20 | 14.40 | 52.00 | 29.20 | 28.70 | 14.40 | 14.80 | 51.20 | 43.14 | 30.89 | 54.79 |
| LLaVA-OV-7B[†] [3] | 1 FPS | 76.84 | 77.17 | 82.60 | 75.25 | 64.15 | 64.17 | 75.00 | 61.38 | 61.19 | 46.11 | 68.39 | 24.40 | 12.00 | 32.40 | 37.60 | 26.60 | 20.00 | 19.60 | 34.40 | 52.94 | 31.74 | 50.96 |
| + *Stream-IT* | 1 FPS | 82.29 | 72.44 | 92.09 | 80.86 | 71.07 | 74.46 | 75.00 | 62.20 | 70.26 | 28.50 | 70.92 | 20.80 | 13.20 | 43.60 | 27.60 | 26.30 | 20.40 | 17.60 | 41.60 | 37.26 | 29.21 | 51.73 |
| Qwen2-VL-7B[†] [2] | 1 FPS | 80.38 | 78.74 | 83.22 | 79.86 | 74.21 | 69.47 | 77.78 | 63.41 | 69.97 | 43.01 | 72.01 | 32.00 | 15.20 | 39.60 | 38.40 | 31.30 | 25.20 | 21.20 | 33.20 | 66.67 | 36.57 | 55.09 |
| + *Stream-IT* | 1 FPS | 84.74 | 82.68 | 88.92 | 89.77 | 77.36 | 85.36 | 84.26 | 69.92 | 71.67 | 35.75 | 77.04 | 18.00 | 13.20 | 43.60 | 21.60 | 24.10 | 14.00 | 17.20 | 48.00 | 50.98 | 32.55 | 55.39 |

Table 11: Full results on Streaming-Bench. [†] means models under *StreamBridge* framework

# I Pseudo Code of the Round-Decayed Compression in a PyTorch-like Style

```python
def Round_Decayed_Compression (inputs_embeds, max_len, token_per_frame):
    '''
    inputs_embeds: [1, seq_len, dim], interleaved embeddings of video and text;
    max_len: the predefined maximum sequence length of inputs_embeds;
    token_per_frame: the number of tokens per frame;
    '''

    # compress_target_num is the number of tokens that need to be compressed,
    # should be integer multiples of token_per_frame
    redudant_frame_num = int((inputs_embeds.shape[1] - max_len)/token_per_frame) + 1
    compress_target_num = token_per_frame * redudant_frame_num

    # split inputs_embeds into image_embeds and text_embeds by round,
    # e.g., image_embeds[i] is the visual tokens of the i-th round,
    # and len(image_embeds) == len(text_embeds) == number of rounds;
    image_embeds, text_embeds = split_image_and_text(inputs_embeds)
    new_inputs_embeds = []

    # compress visual tokens round by round
    for round_idx in range(len(image_embeds)):
        current_image_embeds = image_embeds[round_idx]
        current_text_embeds = text_embeds[round_idx]
        if compress_target_num > 0 and current_image_embeds.shape[1] >=
            token_per_frame*2:

            """
            compress current_image_embeds into [1, token_per_frame, dim];
            """
            if current_image_embeds.shape[1] <= compress_target_num +
                token_per_frame:
                current_frame_num = current_image_embeds.shape[1] // token_per_frame
                current_image_embeds = current_image_embeds.reshape(1,
                    current_frame_num, token_per_frame,
                    current_image_embeds.shape[-1])
                current_image_embeds = current_image_embeds.mean(dim=1)
                compress_target_num -= (current_frame_num-1)*token_per_frame

            """
            compress current_image_embeds's first compress_target_num +
                token_per_frame tokens into [1, token_per_frame, dim], and reserve
                the rest tokens;
            """
            else:
                pre_image_embeds = current_image_embeds[:,
                    :compress_target_num+token_per_frame, :]
                pre_frame_num = pre_image_embeds.shape[1]//token_per_frame
                pre_image_embeds = pre_image_embeds.reshape(1,
                    compress_target_num//token_per_frame + 1, token_per_frame,
                    current_image_embeds.shape[-1])
                pre_image_embeds = pre_image_embeds.mean(dim=1)

                post_image_embeds = current_image_embeds[:,
                    compress_target_num+token_per_frame:, :]
                compress_target_num -= (pre_frame_num-1)*token_per_frame
                current_image_embeds = torch.cat([pre_image_embeds,
                    post_image_embeds], dim=1)

        new_inputs_embeds.append(current_image_embeds)
        new_inputs_embeds.append(current_text_embeds)

    return torch.cat(new_inputs_embeds, dim=1)
```

