# OpenReview forum: "StreamBridge: Turning Your Offline Video Large Language Model into a Proactive Streaming Assistant"
_NeurIPS.cc/2025/Conference — NeurIPS 2025 poster_

### Official Review · Reviewer_neQ8 · 2025-06-21

**Clarity:** 3
**Significance:** 3
**Originality:** 3
**Rating:** 4
**Confidence:** 4

**Summary:**

This paper proposes StreamBridge that transforms offline Video-LLMs into streaming-capable models. StreamBridge designs a memory buffer to efficiently process video inputs and utilizes a lightweight activation model for proactive model responses. Additionally, proposed Stream-IT improves further models for streaming settings. StreamBridge demonstrates its effectiveness on both streaming and offline settings on various benchmarks.

**Questions:**

See the Weaknesses.

**Ethical Concerns:**

["NO or VERY MINOR ethics concerns only"]

**Final Justification:**

The authors have addressed the concerns. I will keep a positive score for the paper.

**Limitations:**

Yes.

**Quality:**

4

**Strengths And Weaknesses:**

**Strengths**

- The motivation is clear and straightforward; Existing Video-LLMs mainly explore offline video understanding settings and may not be capable of streaming settings. StreamBride is proposed to fill this gap.

- The modeling is reasonable; A memory buffer with round-decayed compression is able to process video inputs efficiently, and the activation model is highly flexible.

- The paper provides an extensive experiment; StreamBridge has been evaluated on both streaming and general video understanding tasks. With detailed analyses, StreamBridge demonstrates noticeable improvements.


**Weaknesses**

- When the embedding length of the memory buffer exceeds MaxLen, how exactly are earlier frame embeddings pooled? The paper mentions 1 FPS video sampling, but it's unclear whether mean pooling is applied only to the first two frames or recursively to earlier segments. A more detailed explanation or visualization would clarify the compression behavior over long sequences.
- Line 141 suggests the model retains recent frames preferentially. What empirical evidence supports the assumption that recent visual context is more informative than earlier frames?
- I have some concerns about the activation model. This module resembles a frame sampling strategy that selects the most likely informative frames [1] for a given question and answer. Given that it relies on representations rather than generative capabilities, is an MLLM necessary for this module? Additionally, how do the authors decide the range of P, i.e., 0 ~ 50%? Are all answers in the dataset truly answerable with the last P% of video frames?
- While Stream-IT improves performance across tasks, it significantly underperforms the base models on the Counting (CT) task in Streaming-Bench Real-Time. What might explain this drop? Does the memory update mechanism or frame prioritization bias affect temporal accumulation required for counting?

**References**

[1] Revisiting the “Video” in Video-Language Understanding, CVPR 2022

---

> ### Author Rebuttal · Authors · 2025-07-28
>
> We thank Reviewer neQ8 for the positive evaluation and detailed feedback. We are encouraged by the acknowledgment of our clear motivation, the reasonableness of our modeling choices, and the extensive experimental validation. We address your concerns below.
>
> **(1) Clarification on how early frames are compressed.**
> We appreciate the opportunity to clarify the round-decayed compression mechanism. When the total length of visual tokens exceeds the predefined **$\textbf{MaxLen}$**, we apply recursive average pooling starting from the earliest round. This process ensures that compression proceeds round-by-round, prioritizing the preservation of more recent frames.
>
> To illustrate this, consider a simple example where **$\textbf{MaxLen}$ = 200**, each frame contributes **64** tokens (i.e., **tokens-per-frame = 64**), and the current memory buffer consists of the following sequence:
>
> **Memory buffer**: frame$_1$, frame$_2$, frame$_3$, Q$_1$, A$_1$, frame$_4$, frame$_5$, Q$_2$
>
> Here, frame$_1$–frame$_3$ belong to the **first round**, and frame$_4$–frame$_5$ belong to the **second round**. The total number of visual tokens is **64 × 5 = 320**, which exceeds **$\textbf{MaxLen}$ = 200**. Thus, at least **320 - 200 = 120** tokens must be removed. Since our average pooling reduces frame tokens in units of **tokens-per-frame = 64**, we round up the target to **128** tokens. To achieve this, we apply **average pooling** over the first three frames in the **first round**, collapsing **64 × 3 = 192** tokens into **64** tokens by summing them point-wise and dividing by 3. This reduces the total by **192 - 64 = 128** tokens, bringing the sequence within the **$\textbf{MaxLen}$** constraint, without compressing visual tokens in the 2nd round.
>
> Now, consider a more constrained case with **$\textbf{MaxLen}$ = 150**. We would then need to remove **320 - 150 = 170** tokens, rounded up to **192**. As before, pooling the **first round (frame$_1$–frame$_3$)** reduces **128** tokens. However, this is insufficient, so we continue to the second round, pooling frame$_4$ and frame$_5$ into a single frame representation, further reducing **64** tokens. This results in a total reduction of **128 + 64 = 192** tokens, satisfying the compression requirement.
>
> This **round-wise, progressive** pooling strategy ensures that older rounds are compressed more aggressively while preserving recent, fine-grained visual details, aligning with our round-decay design philosophy. This process is also described with the **pseudo code in Appendix H**, and we will add more explanations in the final version.
>
> **(2) Justification for "recent frames are more informative"**
> We appreciate the reviewer’s insightful question. The design choice to preferentially retain recent frames is primarily motivated by the nature of streaming video understanding, where decisions are made **incrementally and based on the most recent context**. In such scenarios, recent visual information is often more indicative of the current event or interaction state. For instance, in tasks like action detection or dense captioning, models must react to what is happening **now** rather than rely on distant historical context.
>
> To validate this empirically, we conducted compression ablation studies as reported in **Table 4**, where we compared our round-decayed compression with alternative strategies, including uniform pooling and FIFO truncation. The round-decay method, which progressively compresses earlier rounds while retaining fine-grained recent inputs, consistently outperformed other approaches across all streaming tasks. This confirms that recent frames indeed carry more task-relevant information in the streaming setup.
>
> **(3) Necessity of Using an MLLM for the Activation Model and the Choice of P%.**
> Our goal is not merely to identify informative frames, but to decide **when** the model has accumulated **enough multimodal evidence** to confidently produce a response. This is inherently a **causal, streaming process**, where the model **incrementally** observes interleaved video-text inputs over time and must decide whether sufficient understanding has been achieved at each moment. An MLLM is well-suited for this role, as it is trained to perform **causal prediction** (i.e., next token prediction) over sequences of image/video and text. This makes it fundamentally different from offline frame importance estimators (e.g., [1]), which do not model sequential decision-making in a streaming context. Additionally, MLLM is pre-trained on large-scale multimodal datasets and thus benefits from strong **generalization** across diverse visual concepts and task formats. This enables the model to adapt to a wide variety of query types and video domains, including those unseen during activation model fine-tuning.
>
> Regarding the range of P%, we dynamically sample values from **0–50%** to determine which portion of the video contains answer-relevant evidence. This strategy encourages the model to generate responses only **after observing sufficient context**(i.e., **the first (1–P)%** of the video). On average, this means the model responds after seeing around **75%** of the video content. This reflects a realistic streaming behavior, where answers are typically derived from progressively revealed context, and discourages too early activations.
>
> **(4) Counting task underperformance.**
> We agree with the reviewer that the performance drop likely stems from the fact that counting tasks inherently require high temporal fidelity and full preservation of fine-grained visual details across time, while our round-decayed compression strategy is designed to prioritize temporal efficiency and compact historical context, which can occasionally lead to information loss in dense, detail-sensitive tasks like counting. To mitigate this, we plan to augment Stream-IT with more counting-style tasks in future iterations, encouraging the model to retain finer temporal granularity when necessary.
>
> [1] Revisiting the “Video” in Video-Language Understanding, CVPR 2022

---

> > ### Comment · Reviewer_neQ8 · 2025-08-01
> > **Rebuttal Comment**
> >
> > I carefully read the reviews from other reviewers as well as the authors’ rebuttal. I appreciate the authors’ responses to my concerns. However, it appears that other reviewers also raised concerns regarding the model’s reliance on recent frames and activation behavior. I would like to see how the model (along with the proposed methods) is affected by the MAXLEN **L**. If the model primarily focuses on recent frames, would reducing **L** not significantly affect performance? Can the authors comment on this?

---

> > > ### Author Response · Authors · 2025-08-02
> > >
> > > We thank the reviewer for the thoughtful follow-up. To better understand the impact of **MaxLen**, we conducted additional ablation studies using the Qwen2-VL-StreamBridge model with 1 FPS sampling, varying **MaxLen** from **4k** to **32k**:
> > >
> > > | MaxLen (L) | OVO-Bench Real-Time (Acc %) | VideoMME (Acc %) |
> > > |------------|-----------------------------|------------------|
> > > | 4k         | 70.49                        | 61.7             |
> > > | 8k         | 70.89                        | 63.6             |
> > > | 16k        | 71.30                        | 64.4             |
> > > | 32k        | 71.16                        | 64.7             |
> > >
> > > From the table, we observe the following:
> > >
> > > **For streaming tasks (e.g., OVO-Bench Real-Time):**
> > >
> > > 1. Model performance remains relatively stable across varying **MaxLen** values, ranging from 70.49% to 71.30%.
> > >
> > > 2. Accuracy peaks at **16k** and slightly declines at **32k**, suggesting that further increasing the memory budget yields diminishing returns.
> > >
> > > 3. **This supports our design assumption**: in streaming scenarios, models primarily rely on recent context, and older frames can be compressed without significant performance loss. Our round-decayed compression strategy ensures that recent frames are retained at high resolution while historical context is stored compactly, yielding efficient inference under limited memory budgets.
> > >
> > > **For offline tasks (e.g., VideoMME):**
> > >
> > > 1. Accuracy improves consistently as **MaxLen** increases, from 61.7% at **4k** to 64.7% at **32k**.
> > >
> > > 2. This means that offline tasks benefit more from retaining the full temporal context and more uncompressed video tokens, especially for long videos that require detailed long-range understanding. It also highlights that while our method is optimized for streaming, it can **scale up MaxLen to handle offline tasks** effectively when more memory is available (e.g., when **MaxLen ≥ 16k**).
> > >
> > > Overall, these findings highlight the adaptability of StreamBridge: our round-decayed compression is particularly effective for streaming tasks, where recent context is most informative and performance remains stable even under limited memory budgets with lower **MaxLen**. For offline tasks that require comprehensive long-term understanding, increasing **MaxLen** leads to noticeable gains. This demonstrates that StreamBridge can flexibly balance efficiency and performance across both streaming and offline settings by adjusting the memory budget accordingly, and we set **MaxLen = 16k** to strike a good balance between them across most tasks.
> > >
> > > We will include these results and further discussions in the final paper to clarify the role and trade-offs of **MaxLen** in our framework. We appreciate the reviewer’s suggestion, which has helped us strengthen our empirical analysis.

---

> > > > ### Comment · Reviewer_neQ8 · 2025-08-03
> > > > **Rebuttal Comment**
> > > >
> > > > Thanks for the detailed answer. I have no further questions.

---

### Official Review · Reviewer_ySNS · 2025-06-30

**Clarity:** 3
**Significance:** 3
**Originality:** 3
**Rating:** 4
**Confidence:** 3

**Summary:**

This paper proposed a StreamBridge framework to transform offline video LLMs into streaming-capable models. A memory buffer and a decoupled, lightweight modules are incorporated into existing video LLMs to address fundamental challenges. Besides, a new benchmark is proposed for streaming video understanding tasks. Experiments demonstrate the effectiveness of the proposed method in various video understanding tasks.

**Questions:**

Overall, this paper exhibits high-quality writing, solid methodology, and sufficient experiments. I have some minor concerns listed in **weaknesses**.

**Ethical Concerns:**

["NO or VERY MINOR ethics concerns only"]

**Final Justification:**

The authors rebuttal has addressed my initial concerns. I will keep positive score for this paper.

**Limitations:**

Yes

**Quality:**

3

**Strengths And Weaknesses:**

**Strengths**
- The paper writing is good. The motivations are clearly explained, and all the modules are analysed sufficiently.

- The experimental results look good. In most cases, the introduction of StreamBridge brings consistent improvement to various baseline video LLMs.

- A new benchmark is proposed that contains long-context, interleaved video-text instruction tuning, and includes diverse tasks and realistic multi-turn formats.

**Weaknesses**

- The overall performance is easily influenced by the choice of activation models and activation threshold $\alpha$. The authors chose the LLaVA-OV-0.5B as this activation model. Please explain the reason and discuss its influence. From Figure 5, it appears that the variation of $\alpha$ has a significant impact on the overall performance. Does this effect also happen in other benchmarks?

- In Table 2, two results show performance degradation. These results don't match the statement ''we also verify
that it does not downgrade the base model’s performance on the standard offline video task in L256.

- In Section 3.2.3, the authors used a heuristic sampling strategy that assigns the last P% frames as positive. It is necessary to discuss the effect of P% under different settings (e.g., set a fixed P% for each sample).

---

> ### Author Rebuttal · Authors · 2025-07-28
>
> We thank Reviewer ySNS for the encouraging feedback. We are pleased that the reviewer found the writing clear, the experimental results looking good, and the effectiveness of our constructed dataset. Below, we address your comments.
>
> **(1) Choice of LLaVA-OV-0.5B as the activation model.**
> We chose LLaVA-OV-0.5B for its **lightweight** architecture and **fast** inference speed, making it well-suited as a concurrent decision module that introduces minimal computational overhead. We agree with the reviewer that the activation threshold $\alpha$ plays a role in response frequency. In practice, we empirically set $\alpha$ within the range of **[0.3, 0.4]**, which we found to strike a good balance between precision and responsiveness across most tasks. This design also enables flexible control depending on downstream application requirements.
>
> **(2) Performance drop in Table 2 (Offline Tasks).**
> Thank you for pointing this out. The observed performance drops (e.g., -2.0 on LongVideoBench and -2.6 on MVBench) are isolated cases out of nearly 21 benchmark-model combinations. We will clarify these exceptions and emphasize the broader non-destructive trend. We also commit to using more accurate and objective language in the final version of the paper.
>
> **(3) Impact of the P% heuristic in activation model.**
> We appreciate the reviewer’s thoughtful observation. In our design, we dynamically sample P% between **0–50%** to simulate varied activation delays. This range encourages the model to avoid generating premature responses in the very early portion of a video, where insufficient context would make meaningful responses infeasible. The model is thus trained to wait until it has observed at least **(1–P)%** of the video before considering a response. On average, this means the model typically responds after viewing around **75%** of the video, which aligns with the intuition that robust understanding often requires seeing most of the content. Moreover, dynamic sampling introduces beneficial randomness, helping the model **avoid learning shortcut patterns**, such as always responding at fixed positions and promotes stronger generalization across diverse scenarios.
>
> To further validate this, we compare two settings on ET-Bench: one with **a fixed P = 25%** and another with **P uniformly sampled from 0–50%**:
>
> | Range of P                  | TVG$_{F1}$ | TAL$_{F1}$ | DVC$_{F1}$ | SLC$_{F1}$ |
> |-----------------------------------------|------------|------------|------------|------------|
> | P = 25%                               | 34.1       | 21.6       | 33.9       | 18.4       |
> | P ~ $\mathcal{U}$(0, 50\%) | 34.3       | 24.3       | 37.8       | 22.5       |
>
> As shown, dynamic sampling of P% leads to improved performance and generalization across tasks, supporting the effectiveness of this design choice.

---

> > ### Comment · Reviewer_ySNS · 2025-08-04
> > **Additional questions**
> >
> > The authors' rebuttal has addressed most of my concerns. I still have one question about the choice of activate model. The authors claimed that the choice of LLaVA-OV-0.5B is due to its efficiency for the whole model. However, from the comparison results in Table 2, the proposed method didn't show significant superiority in inference speed. I think adding a detailed time analysis on each component of the proposed method is helpful to address this concerns. Besides,  I am also curious about the time degradation if the smaller LLaVA-OV-0.5B is replaced by a larger backbone (no need to report the accuracy, just report time).

---

> ### Author Response · Authors · 2025-08-04
>
> We thank the reviewer for the thoughtful follow-up.
>
> To further support our efficiency claim, we conducted a breakdown of the runtime and memory usage of the two key components in our framework: (1) the main Video-LLM (Qwen2-VL-7B) and (2) the activation model (LLaVA-OV-0.5B vs. LLaVA-OV-7B). All experiments were performed on the subset of ET-Bench using 1 FPS input with a single NVIDIA A6000 (48GB) GPU. We report the average inference time and memory usage:
>
> | Model                             | Runtime (s) | Memory Usage (GB) |
> |----------------------------------|-------------|-------------------|
> | Main Video-LLM: Qwen2-VL-7B      | 2.79        | 17.7              |
> | Activation Model: LLaVA-OV-0.5B  | 0.35        | 1.67              |
> | Activation Model: LLaVA-OV-7B    | 1.87        | 15.9              |
>
> As shown, compared with the main Video-LLM (Qwen2-VL-7B), the lightweight activation model (LLaVA-OV-0.5B) only contributes less than **12.5%** of the runtime (0.35s vs 2.79s) and **9.4%** of the memory usage (1.67GB vs 17.7GB), validating our design choice of using LLaVA-OV-0.5B for efficiency. In contrast, replacing it with a larger backbone (LLaVA-OV-7B) results in approximately a **9.5× increase in memory usage** and significantly longer inference time, making it impractical for real-time streaming scenarios.
>
> We appreciate the reviewer’s insightful suggestion, which helped strengthen the justification of our architectural choices. We will incorporate all of these into our paper revision.

---

> > ### Comment · Reviewer_ySNS · 2025-08-05
> >
> > Thanks for the authors' explanations. I have no questions, and I keep positive assessment for this paper.

---

### Official Review · Reviewer_ioHp · 2025-07-02

**Clarity:** 3
**Significance:** 3
**Originality:** 2
**Rating:** 4
**Confidence:** 4

**Summary:**

This paper introduces StreamBridge, a framework designed to transform offline Video-LLMs into streaming-capable models. The key challenges addressed are (1) enabling multi-turn real-time understanding and (2) implementing proactive response mechanisms. StreamBridge introduces a memory buffer and a round-decayed compression strategy to manage streaming inputs efficiently. It also constructs a large-scale dataset for streaming video understanding. The approach is evaluated on benchmarks like Streaming-Bench and OVO-Bench, demonstrating competitive performance against existing models.

**Questions:**

See the weaknesses part above.

**Ethical Concerns:**

["NO or VERY MINOR ethics concerns only"]

**Final Justification:**

The rebuttal has adequately addressed my concerns. While the paper is technically sound, its novelty remains limited (as noted in Q1). Nevertheless, I am raising my score and am now inclined toward a "Borderline accept" recommendation.

**Limitations:**

Yes

**Quality:**

3

**Strengths And Weaknesses:**

### Strengths
1. StreamBridge addresses a critical gap in adapting offline Video-LLMs to real-time streaming scenarios, which has broad applications in video analysis and interactive applications.
2. The paper presents extensive experiments across multiple streaming benchmarks (e.g., OVO-Bench, Streaming-Bench) and ablation studies.
3. The newly introduced Stream-IT dataset provides valuable support for training streaming video understanding models, promoting future research in this emerging area.

### Weaknesses
1. While effective, the Memory Buffer and Round-Decayed Compression strategies have been explored in prior work (e.g., LSTR [1]) in different contexts. This diminishes the perceived technical originality of the approach.
2. The compression strategy assumes that recent frames are more relevant to incoming queries. However, this assumption may not hold universally and could be influenced by benchmark biases. A deeper investigation is needed to validate its general effectiveness. For example, conducting an ablation study where the question is intentionally delayed by varying time intervals, simulating scenarios in which a query relates to a past segment rather than the most recent visual input.
3. The paper refers to `Qwen2-VL†`’s +2.97 improvement on Streaming-Bench as a “notable improvement,” while characterizing `LLaVA-OV†`’s -2.38 and -2.73 drops on OVO-Bench and Streaming-Bench as “slight performance drops” (L249-252). This language appears inconsistent and subjective.
4. The use of a lightweight model (`LLaVA-OV-0.5B`) and aggressive token compression in the Activation Model raises concerns about its accuracy. However, no quantitative evaluation or ablation study is provided to validate its effectiveness. An ablation study would strengthen confidence in this component.
5. The related work section would benefit from additional discussions with more relevant prior works, such as [2–4], to better situate the proposed approach within the context of streaming video understanding.

---
[1] Xu, Mingze, et al. "Long short-term transformer for online action detection." Advances in Neural Information Processing Systems 34 (2021): 1086-1099.

[2] Qian, Rui, et al. "Streaming long video understanding with large language models." Advances in Neural Information Processing Systems 37 (2024): 119336-119360.

[3] Di, Shangzhe, et al. "Streaming video question-answering with in-context video kv-cache retrieval." arXiv preprint arXiv:2503.00540 (2025).

[4] Xiong, Haomiao, et al. "Streaming Video Understanding and Multi-round Interaction with Memory-enhanced Knowledge." arXiv preprint arXiv:2501.13468 (2025).

---

> ### Author Rebuttal · Authors · 2025-07-28
>
> We thank Reviewer ioHp for the thoughtful review and valuable comments. We are glad that the reviewer acknowledged StreamBridge's effort to fill a critical gap in adapting offline Video-LLMs to real-time streaming scenarios, and appreciated our extensive benchmark coverage and the introduction of the Stream-IT dataset. We address your suggestions below.
>
> **(1) Novelty of memory buffer and round-decayed compression**
>
> We clarify that while memory mechanisms are not entirely new (e.g., LSTR), our method is specifically tailored to **multi-turn streaming** scenarios with a fixed token budget. Unlike LSTR, which maintains separate long- and short-term memories limited to only visual tokens, our memory buffer stores **interleaved video-text** tokens, enabling richer and more coherent contextual understanding and multi-turn dialogues for multimodal LLMs.
>
> Moreover, our round-decayed token compression is a lightweight and **training-free** strategy for inference only. In contrast, LSTR introduces an additional LSTR Decoder that requires training to extract information from long-term memories via cross-attention. Our method avoids such complexity, instead merging older visual tokens through simple average pooling while preserving recent context, offering both efficiency and adaptability without additional training overhead.
>
> **(2) Assumption that recent frames are more relevant**
>
> We acknowledge that this assumption may not always hold in **non-streaming** scenarios where earlier frames can contain critical information. To address this, our round-decayed compression strategy does **not discard older content**; instead, it **pools** earlier video tokens to retain coarse-grained historical context. This design is validated in **Table 4** and discussed in Lines 294–303. Furthermore, the memory budget can be adjusted via the **MaxLen** parameter (defined in Line 137). For instance, when **MaxLen = 32k**, the model can store up to **15 minutes** of video at 1 FPS without requiring additional compression, ensuring full retention of video information in low-latency or short-form scenarios.
>
> Moreover, standard offline video benchmarks such as **VideoMME** and **LongVideoBench**, whose questions are posed at the end of the video, inherently include scenarios where "the question is intentionally delayed by varying time intervals, simulating scenarios in which a query relates to a past segment rather than the most recent visual input." As shown in **Table 2**, our approach maintains comparable performance on these benchmarks, demonstrating StreamBridge's generalization that retains relevant historical information and is not biased toward only recent context.
>
> **(3) Inconsistent language in result interpretation**
>
> Thank you for pointing this out. We will revise the phrasing to ensure consistency in describing all performance changes across models.
>
> **(4) Activation model's ablation**
>
> We agree and appreciate the suggestion. In our current setting, we use **16 tokens per frame** for the activation model. Below, we provide additional quantitative results on **ET-Bench**, exploring the impact of token granularity. We report only the **F1 score**, which reflects the precision of the activation model’s response timing:
>
> **Table: Activation Token Granularity on ET-Bench**
>
> | Tokens per Frame | TVG-F1| TAL-F1| DVC-F1| SLC-F1 |
> |------------------|------------------|------------------|------------------|------------------|
> | 8                | 30.1             | 19.3             | 33.8             | 19.1             |
> | 16               | 34.3             | 24.3             | 37.8             | 22.5             |
> | 32               | 35.1             | 23.1             | 38.3             | 21.9             |
> | 64               | 34.9             | 25.3             | 38.8             | 22.8             |
>
> As shown, increasing the number of tokens per frame does not significantly improve performance. This is expected, as the activation model is designed to perform **binary classification** (i.e., whether to trigger a response), which requires less fine-grained semantic detail than generative tasks. Our default setting of a small MLLM and 16 tokens per frame achieves a good balance between efficiency and accuracy.
>
> **(5) Related work coverage**
>
> Thank you for the references. We will expand the related work section to include [1, 2, 3], and incorporate a discussion of them to provide a more complete context for our work.
>
> [1] Qian, Rui, et al. *"Streaming long video understanding with large language models."* NeurIPS 2024.
>
> [2] Di, Shangzhe, et al. *"Streaming video question-answering with in-context video kv-cache retrieval."* arXiv:2503.00540, 2025.
>
> [3] Xiong, Haomiao, et al. *"Streaming Video Understanding and Multi-round Interaction with Memory-enhanced Knowledge."* arXiv:2501.13468, 2025.

---

> > ### Comment · Reviewer_ioHp · 2025-08-03
> >
> > Thanks for your responses, which have addressed some of my concerns (Q1-Q3). However, the following issues still require clarification or further validation:
> >
> > - Regarding Q4: The current ablation analysis appears incomplete. Specifically, how does using a larger activation model impact the results?
> > - Regarding Q5: While the authors promise to add discussion in the final draft, I find it disappointing that no substantive discussion was provided in this rebuttal despite having ample space to do so.

---

> ### Author Response · Authors · 2025-08-04
>
> **Impact of using a larger activation model**
>
> We appreciate the reviewer’s follow-up. In addition to varying the token granularity, we have also conducted experiments using a larger activation model (LLaVA-OV-7B) to study the effect of model scale. We compare its performance against our default lightweight model (LLaVA-OV-0.5B) on ET-Bench:
>
> | Activation Model | TVG-F1| TAL-F1 | DVC-F1 | SLC-F1 | Memory Usage (GB) |
> |------------------|------------------|------------------|------------------|------------------|------------------|
> | LLaVA-OV-0.5B     | 34.3             | 24.3             | 37.8             | 22.5             |  1.67   |
> | LLaVA-OV-7B        | 36.5             | 27.1             | 39.6             | 24.0             |  15.9   |
>
> While LLaVA-OV-7B offers a performance gain (about 2.1 F1 improvement on average), we find that the benefit does not justify the significantly increased computational cost (1.67GB vs 15.9GB), particularly in streaming scenarios. Hence, we opt for the smaller model as a better trade-off between efficiency and effectiveness.
>
> **Discussion of Related Work**
>
> Thank you again for the suggestion. Below is a brief discussion on how our work relates to the suggested references:
>
> To address the challenge of long-context understanding in streaming video, several memory and retrieval mechanisms have been proposed. For instance, **ReKV** [1] introduces a training-free framework that stores and retrieves the Key-Value (KV) caches of processed frames, enabling offline models to answer user queries efficiently by reloading only the most relevant context. Besides, **VideoStreaming** [2] employs a memory-propagated encoding architecture where a condensed representation of the preceding clip serves as historical context for encoding the next, combined with an adaptive selection of memories for question-answering. Moreover, **StreamChat** [3] proposes a hierarchical memory system comprising short-term, long-term, and dialogue components to facilitate complex streaming interactions, and also contributes the **StreamBench** benchmark for evaluating diverse streaming scenarios. While these methods effectively advance long-context retention for reactive question-answering, StreamBridge differs by introducing a round-decayed compression strategy specifically tailored for multi-turn real-time interactions, which efficiently prunes redundant historical tokens while preserving recent context with high fidelity. Moreover, StreamBridge introduces a decoupled, lightweight activation model. This plug-and-play component operates in parallel with the main Video-LLM, enabling continuous proactive responses. These designs, supported by our dedicated Stream-IT dataset, effectively transform general-purpose offline models into versatile and proactive streaming assistants without compromising their core performance.
>
> We will incorporate these discussions in the final version to better position our work within the broader landscape of streaming video understanding.
>
> [1] Di, Shangzhe, et al. "Streaming video question-answering with in-context video kv-cache retrieval."
> [2] Qian, Rui, et al. "Streaming long video understanding with large language models."
> [3] Xiong, Haomiao, et al. "Streaming Video Understanding and Multi-round Interaction with Memory-enhanced Knowledge."

---

### Official Review · Reviewer_Qdek · 2025-07-03

**Clarity:** 3
**Significance:** 3
**Originality:** 2
**Rating:** 4
**Confidence:** 3

**Summary:**

This paper introduces StreamBridge, a framework designed to adapt existing offline Video-LLMs for proactive, real-time streaming video understanding. The authors identify two primary challenges in this domain: the lack of multi-turn real-time understanding and the absence of proactive response mechanisms in current models. To address these, StreamBridge proposes three key components: 1. A memory buffer to manage the history of interleaved video and text inputs. 2. A round-decayed compression strategy to efficiently handle long-context inputs by merging older visual tokens while preserving recent ones. 3. A decoupled, plug-and-play activation model that runs in parallel to the main Video-LLM to decide when a response should be generated, enabling proactive behavior without interfering with the main model's architecture.

Furthermore, the authors contribute Stream-IT, a large-scale instruction-tuning dataset specifically created for streaming scenarios, featuring interleaved video-text sequences and a novel synthetic component, StreamingQA-120K. The extensive experiments demonstrate that models adapted with StreamBridge and fine-tuned on Stream-IT not only achieve state-of-the-art performance on streaming benchmarks, outperforming proprietary models like GPT-4o and Gemini 1.5 Pro, but also maintain or improve their capabilities on standard offline video understanding tasks.

**Questions:**

See Weaknesses

**Ethical Concerns:**

["NO or VERY MINOR ethics concerns only"]

**Final Justification:**

My concerns have been addressed. I would like to maintain my scores (Borderline accept).

**Limitations:**

See Weaknesses

**Quality:**

3

**Strengths And Weaknesses:**

1. Strengths

1.1 Effective framework design

The proposed StreamBridge framework is simple yet powerful. The decision to use a decoupled activation model is particularly elegant. It avoids the optimization conflicts and performance degradation that can arise from tightly integrating activation mechanisms into the main model's architecture, a known issue in prior work. This modular, plug-and-play approach enhances flexibility and preserves the base model's core capabilities.

1.2 Resource contribution (Stream-IT Dataset)

The creation and release of the Stream-IT dataset is a major contribution to the community. The methodology for constructing the long-form, multi-turn StreamingQA-120K portion by concatenating semantically related clips and using a powerful LLM (GPT-4o) for QA generation is sound and addresses a clear data gap for research in this area. The data augmentation strategies (Random QA Drop and QA Interval Shift) are thoughtful additions that increase the dataset's diversity and utility for training robust streaming models.

1.3 Comprehensive empirical evaluation

The experiments are thorough and the results are compelling. The authors demonstrate the framework's generalizability by applying it to three different base models (Oryx-1.5, LLaVA-OV, Qwen2-VL). The fact that their fine-tuned models outperform even powerful proprietary systems on challenging streaming benchmarks (OVO-Bench, Streaming-Bench) is a strong result. Crucially, they also validate that this improvement does not come at the cost of performance on general offline tasks, showcasing the non-destructive nature of their adaptation method. The ablation studies effectively justify their design choices, particularly the round-decayed compression strategy and the components of the Stream-IT dataset.

2. Weaknesses

2.1 Generalization of the activation model:

The activation model is trained on a collection of datasets from five specific tasks using predefined prompt templates. How well would this activation model generalize to completely unseen tasks or scenarios where the need for a proactive response is more subtle and not guided by an initial, task-specific prompt? The reliance on templates might limit its true zero-shot proactive capabilities.

2.2 Lack of direct efficiency comparison

The paper reports the operational FPS for StreamBridge and other streaming architectures in the results tables, but it lacks a direct, controlled comparison of end-to-end latency and computational efficiency on the same hardware. Reported FPS from different papers can be influenced by numerous factors and is not a substitute for a rigorous head-to-head benchmark. Such an analysis would provide a much clearer understanding of StreamBridge's practical efficiency relative to methods like VideoLLM-Online or Flash-VStream.

---

> ### Author Rebuttal · Authors · 2025-07-28
>
> We sincerely thank Reviewer Qdek for the insightful and constructive feedback. We are particularly encouraged by the recognition of our modular and elegant StreamBridge framework, the contribution of the Stream-IT dataset, and the comprehensive empirical evaluation. We address your concerns below.
>
> **(1) Generalization of the activation model**
>
> We agree that generalization is a critical aspect of the activation model. To encourage robustness, our model is trained on five diverse tasks (e.g., dense captioning, step localization, action detection, etc.) using a variety of prompt templates and a dynamic P% sampling strategy, which enables the model to learn activation across varied temporal patterns.
>
> To further evaluate its generalization ability, we report results on **NExT-GQA** [1], a grounded VideoQA benchmark characterized by diverse, natural question prompts with annotated timestamps. Unlike template-based tasks like dense video captioning, NExT-GQA requires the model to trigger a response precisely when the visual evidence for answering the question appears, presenting a strong challenge for generalization.
>
> **Table: Activation Model Generalization on NExT-GQA**
>
> |               | Absolute (s) | Relative (%) |
> |---------------|---------------|----------------|
> | Activation Delay | 3.24        | 7.7            |
>
> Here in the table, **Absolute** denotes the average time gap (in seconds) between the activation timestamp and the annotated ground-truth response timestamp. **Relative** refers to this gap as a percentage of the total video duration. These results demonstrate that the activation model can proactively answer the question within a delay of 3.24 seconds or 7.7% of the video duration on average, and generalize well to unseen videos and diverse, open-ended questions, even in scenarios not encountered during training.
>
> [1] Xiao, Junbin, et al. *"Can I trust your answer? Visually grounded video question answering."* CVPR 2024
>
> ---
>
> **(2) Direct efficiency comparison**
>
> We appreciate the reviewer’s suggestion and now provide a direct inference latency comparison between **Qwen2-VL-StreamBridge (ours)**, **VideoLLM-Online**, **Flash-VStream**, and **Dispider**. All models are evaluated on the ActivityNet dense video captioning task using the same hardware (a single NVIDIA A6000-48G GPU) with 1 FPS video sampling:
>
> **Table: Latency and Accuracy Comparison**
>
> | Models                  | ANet-Caption Latency (s) | OVO-Bench Real-Time Acc (%) |
> |-------------------------|---------------------------|-----------------------------|
> | Qwen2-VL-StreamBridge (ours) | 1.52                      | 71.30                       |
> | Dispider                | 2.47                      | 54.55                       |
> | VideoLLM-Online         | 0.89                      | 20.79                       |
> | Flash-VStream           | 0.95                      | 28.37                       |
>
> From the results, we observe that our framework is more efficient than Dispider, while slower than VideoLLM-Online and Flash-VStream. This trade-off is expected: both VideoLLM-Online and Flash-VStream rely on highly aggressive compression schemes **(1–10 tokens per frame)**, which significantly reduce latency but also lead to substantial information loss. In contrast, StreamBridge maintains a richer representation per frame and achieves significantly stronger results on challenging benchmarks such as **OVO-Bench** and **VideoMME**, highlighting a better balance between efficiency and accuracy.

---

### Decision · Program_Chairs · 2025-09-17

**Decision:**

Accept (poster)

**Comment:**

The authors propose StreamBridge, a framework to convert offline Video-LLMs into streaming-capable models and Stream-IT, a large-scale dataset for streaming video understanding. The contributions of this paper include:
1.  Adapting offline Video-LLMs to real-time streaming scenarios is an important research redirection with broad applications.
2. The authors propose a StreamBridge pipeline based on a memory buffer and a decoupled activation model, which is simple yet powerful on various streaming video understanding benchmarks.
3. The collected StreamingQA-120K dataset is beneficial for the research community for training streaming models.

Below are some of the salient discussion points during the reviewer-author discussion period:
1. Reviewers have concerns about the generalization ability of the activation model, since the activation model is trained on specific data. The authors provide additional results on the NExT-GQA benchmark to demonstrate that the activation model generalizes well.
2. Since the topic of this paper is about streaming video understanding, the authors provide additional metrics (e.g., latency) to demonstrate the computational efficiency.
3. Reviewers have concerns that the authors miss some relevant related work (Reviewer ioHp Q5). The authors promise to add the discussion of these papers in the revised version.
4. The authors provide more implementation details and ablation studies of StreamBridge to address the reviewers' concerns.

Overall, the reviews have been positive, and after rebuttal, all reviewers recommend acceptance, acknowledging that it fills an important gap in the streaming video understanding. The authors are encouraged to incorporate these discussions (related works, implementation details) in the final version.